# Application of Low-Cost Fine Particulate Mass Monitors to Convert Satellite Aerosol Optical Depth to Surface Concentrations in North America and Africa

Carl Malings[1,2,3], Daniel M. Westervelt[4], Aliaksei Hauryliuk[5], Albert A. Presto[5], Andrew Grieshop[6],
Ashley Bittner[6], Matthias Beekmann[1,2], R Subramanian[1,2]

[1]OSU-EFLUVE - Observatoire Sciences de l'Univers-Enveloppes Fluides de la Ville à l'Exobiologie, Université Paris-Est-Créteil, CNRS UMS 3563, Ecole Nationale des Ponts et Chaussés, Université de Paris, France
[2]Laboratoire Interuniversitaire des Systèmes Atmosphériques (LISA), UMR 7583, CNRS, Université Paris-Est-Créteil, Université de Paris, Institut Pierre Simon Laplace, Créteil, France
[3]Currently a NASA Postdoctoral Program Fellow, Goddard Space Flight Center, Greenbelt, MD 20771, USA
[4]Lamont-Doherty Earth Observatory, Columbia University, New York, NY, USA
[5]Center for Atmospheric Particle Studies, Carnegie Mellon University, 5000 Forbes Avenue, Pittsburgh, PA 15213, USA
[6]Department of Civil, Construction and Environmental Engineering, North Carolina State University, Raleigh, NC, USA

*Correspondence to*: Carl Malings, R Subramanian (cmalings@alumni.cmu.edu, subu@cmu.edu)

**Abstract.** Low-cost particulate mass sensors provide opportunities to assess air quality at unprecedented spatial and temporal resolutions. Established traditional monitoring networks have limited spatial resolution and are simply absent in many major cities across sub-Saharan Africa (SSA). Satellites provide snapshots of regional air pollution but require ground-truthing. Low-cost monitors can supplement and extend data coverage from these sources worldwide, providing a better overall air quality picture. We investigate the utility of such a multi-source data integration approach using two case studies. First, in Pittsburgh,
Pennsylvania, both traditional monitoring and dense low-cost sensor networks are compared with satellite aerosol optical depth (AOD) data from NASA's MODIS system and a linear conversion factor is developed to convert AOD to surface fine particulate matter mass concentration (as $PM_{2.5}$). With 10 or more ground monitors in Pittsburgh, there is a two-fold reduction in surface $PM_{2.5}$ estimation mean absolute error compared to using only a single ground monitor. Second, we assess the ability of combined regional-scale satellite retrievals and local-scale low-cost sensor measurements to improve surface $PM_{2.5}$
estimation at several urban sites in SSA. In Rwanda, we find that combining local ground monitoring information with satellite data provides a 40% improvement in surface $PM_{2.5}$ estimation accuracy with respect to using low-cost ground monitoring data alone. A linear AOD to surface $PM_{2.5}$ conversion factor developed in Kigali, Rwanda did not generalize well to other parts of SSA, and varied seasonally for the same location, emphasizing the need for ongoing and localized ground-based monitoring, which can be facilitated by low-cost sensors. Overall, we find that combining ground-based low-cost sensor and satellite data,
even without including additional meteorological or land use information, can improve and expand spatio-temporal air quality data coverage especially in data-sparse regions.

## 1 Introduction

Air quality is the single largest environmental risk factor for human health; outdoor air pollution exposure is estimated to have caused about four million premature deaths annually in recent years (WHO, 2016, 2018a). Particulate matter (PM), which represents a mixture of solid and liquid substances suspended in the air, is one of the most commonly tracked and regulated atmospheric pollutants globally (WHO, 2006). Exposure to fine PM is known to have major adverse health impacts (e.g. Schwartz et al., 1996; Pope et al., 2002; Brook et al., 2010). In addition, PM mass concentration is often used as a proxy for overall air quality (WHO, 2018a). PM mass concentration is typically tracked as $PM_{10}$ (total PM mass with diameter below 10 micrometers) and/or $PM_{2.5}$ (total PM mass with diameter below 2.5 micrometers). Even at low concentrations, PM can have significant health impacts (Bell et al., 2007; Apte et al., 2015). These health impacts are especially notable in low-income communities and countries, where they can interact with other socio-economic risk factors (Di et al., 2017; Ren et al., 2018). Sub-Saharan Africa (SSA) in particular is affected by poor air quality, with less than 10% of communities assessed by the WHO meeting recommended air quality guidelines, compared with 18% globally, and 40 to 80% in Europe and North America (WHO, 2018b). This poor air quality manifests in terms of high infant mortality (Heft-Neal et al., 2018), increased risk of chronic respiratory and cardiovascular diseases (Matshidiso Moeti, 2018), and reduced gross domestic product (World Bank, 2016). Industrial development and climate trends will likely only exacerbate this problem in the future (Liousse et al., 2014; UNEP, 2016; Silva et al., 2017; Abel et al., 2018).

Many African countries have among the highest estimated annual average $PM_{10}$ and $PM_{2.5}$ concentrations, yet are also among those with the lowest number of in situ regulatory-grade PM monitoring sites per capita. Fig. 1 shows estimated average annual $PM_{2.5}$ concentrations for various regions of the world versus the density of regulatory-grade monitoring sites in these regions (note that low-cost monitors are not considered), based on information from the Global Health Observatory (GHO). The GHO combines data from multiple sources, including data collected during different years and from sporadic field monitoring campaigns, and it is not necessarily reflective of continuous routine monitoring for all regions (WHO, 2017). This lack of continuous surface monitoring data makes it difficult to answer basic scientific and policy questions related to air quality assessment and mitigation (Petkova et al., 2013; Martin et al., 2019). A major reason for this gap is the high capital and operational costs of traditional ground-based air quality monitoring equipment. Two emerging technologies have the capacity to close this gap: satellite-based air quality monitoring and ground-based low-cost sensor systems.

Satellites are much more expensive than traditional ground-based monitors, but their mobility and unique vantage point allow them to provide near-global coverage. Data from earth-observing satellites can be used to assess air quality in a variety of ways. In particular, aerosol optical depth (AOD) retrievals quantify the absorption and scattering (extinction) of light by the atmosphere and can be related to the concentration of light-absorbing or light-scattering pollutants in the atmosphere. Several factors complicate the relationship between AOD and surface-level particulate matter mass concentrations (Paciorek and Liu, 2009). As a vertically-integrated quantity, AOD is related to total light extinction by a column of atmosphere. The spatial distribution of particulate matter, especially vertical stratification, the presence or absence of plumes aloft, humidity, and the

size and optical properties of particles affect the relationship between AOD and surface concentrations (Kaufman and Fraser, 1983; Liu et al., 2005; Paciorek et al., 2008; Superczynski et al., 2017; Zeng et al., 2018). Cloud cover also makes AOD retrievals impossible; the frequency of cloudy days in an area can therefore make it difficult to establish reliable relationships between AOD and surface PM, although this is not likely to be a concern for the continental US (Christopher and Gupta, 2010; Belle et al., 2017). Changes in surface brightness can also confound this relationship, although this may be less of an issue in

developing countries with higher aerosol levels (Paciorek et al., 2012).

    Nevertheless, early examinations of AOD data from the Moderate Resolution Imaging Spectroradiometer (MODIS) instrument, launched aboard the Terra and Aqua satellites in 1999 and 2002, showed good correlation (e.g. correlation coefficient r about 0.7 for Jefferson County, Alabama in 2002) with surface $PM_{2.5}$ concentrations in the United States, although these relationships varied from region to region (Wang and Christopher, 2003; Engel-Cox et al., 2004). For instance,

correlations between AOD and hourly surface $PM_{2.5}$ were found to vary from an r of 0.6 in the southeastern United States to an r of 0.2 in the southwestern United States during 2005-2006, with root-mean-square errors (RMSE) of about 9 $\mu g/m^3$ for surface $PM_{2.5}$ reconstructed from AOD using linear relationships, with worse results over urban areas (Zhang et al., 2009). Additional studies show broadly similar relationships, with r ranging between about 0.5 and 0.8 in the northeastern United States (e.g. Paciorek and Liu, 2009), with changes in agreement depending on season (Chudnovsky et al., 2013a) and with

better agreement at higher spatial AOD resolution (Chudnovsky et al., 2013b). Using additional covariates, such as land cover, land usage, and meteorological information, can further improve these relationships. In particular, surface $PM_{2.5}$ estimation models combining daily-averaged, 1 km resolution AOD data with meteorological and land use regression variables achieved an agreement (r) with EPA ground-based monitors of up to about 0.95 in the northeastern and 0.9 in the southeastern United States, with a mean absolute error of about 3 $\mu g/m^3$ (Chang et al., 2014; Chudnovsky et al., 2014; Kloog et al., 2014). Methods

incorporating the outputs of chemical transport models (in this case at lower spatial resolutions of 12 km compared to the 1 km AOD resolution, and at daily temporal resolution) can further improve these results (e.g. Murray et al., 2019).

    Models combining satellite AOD data with vertical profiles derived from chemical transport models tend to underestimate surface-level $PM_{2.5}$ outside of Europe and North America, mainly in India and China where ground-based comparison data are available (van Donkelaar et al., 2010, 2015). In China, the r between surface $PM_{2.5}$ estimates derived from satellite AOD,

meteorological, and land use information and measured surface $PM_{2.5}$ was found to be about 0.8, corresponding to a RMSE of about 30 $\mu g/m^3$ (roughly half the mean concentration) in resulting satellite-derived surface concentration estimates (Ma et al., 2014). A method that updates the relationships between AOD and surface $PM_{2.5}$ on a daily basis (Lee et al., 2011) was able to improve these results, increasing r above 0.9 while reducing RMSE to about 20 $\mu g/m^3$ (Han et al., 2018). This method, however, relies on local ground-based measurements to provide the data necessary to perform this daily updating.

Satellites have the potential to provide broad data coverage to previously unmonitored areas such as in SSA. Satellite-based AOD and ground-based AOD agreed well during a recent assessment in West Africa (Ogunjobi and Awoleye, 2019), but an assessment in South Africa found a poor relationship between satellite AOD and surface $PM_{2.5}$, with maxima in the surface concentrations coinciding with minima in the AOD (Hersey et al., 2015). Relationships between AOD and surface $PM_{2.5}$

developed using ground monitoring data elsewhere in the world may not transfer well to SSA, leading to inaccurate

quantification of surface air quality.

Low-cost air quality monitors have much lower purchase and operational costs in contrast to traditional or regulatory-grade monitors (Snyder et al., 2013; Mead et al., 2013). For example, a lower-cost multi-pollutant monitor (measuring gases and PM) costs a few thousand US dollars; single-pollutant PM sensors can cost just a few hundred US dollars. A comparable multi-pollutant suite of traditional air quality monitoring instruments would cost a hundred thousand US dollars or more; a

regulatory-grade PM monitor can cost tens of thousands of US dollar (based on recent manufacturer quotations). This cost reduction is made possible by a combination of lower-cost measurement technologies (such as electrochemical sensors for gases and optical particle detectors for PM) and decreasing costs of battery, data storage, and communications technologies. Much recent research interest has been focused on assessing the performance of these technologies (e.g. AQ-SPEC, 2015, 2017), developing methods for accounting for cross-interference effects in gas sensors (e.g. Cross et al., 2017; Zikova et al.,

2017; Kelly et al., 2017; Zimmerman et al., 2018; Crilley et al., 2018; Malings et al., 2019a) and humidity dependence in optical PM measurement methods (e.g. Malings et al., 2019b) to improve data quality, and demonstrating the utility of these low-cost monitors in various use cases (e.g. Subramanian et al., 2018; Tanzer et al., 2019; Bi et al., 2020). Because of their relatively low cost, these instruments can be deployed more widely than traditional monitoring technologies, enabling measurements in previously unmonitored areas. A trade-off for this increased affordability can be reduced accuracy compared

to traditional air quality monitoring instruments. While there are currently no agreed-upon criteria for assessing low-cost monitor performance (Williams et al., 2019), several schemes suggest tiered rankings ranging from, for example, 20% relative uncertainty for reasonable quantitative measurements to 100% uncertainty for indicative measurements (Allen, 2018); this gives a general sense of the expected performance characteristics of such instruments. In particular, recent testing of two types of such low-cost monitors (which are the types used in this paper) found relative uncertainties on the order of 40% and

correlation coefficient r of 0.7 with regulatory-grade instruments for hourly $PM_{2.5}$ measurements (Malings et al., 2019b). These results are generally consistent with similar studies conducted in a variety of environments and concentration regimes, although relative performance tends to improve at higher concentrations (Kelly et al., 2017; Zheng et al., 2018).

The potential exists to use both satellite and low-cost sensor data together to address the shortcomings of each data source individually and to fill existing data gaps globally. Satellite data provides near-global coverage, but relationships between

AOD and surface $PM_{2.5}$ do not generalize well across regions, and so local ground-based data are needed for establishing conversion factors. Low-cost sensors can provide these local data in areas where existing monitoring networks are sparse or data are sporadically available. The current work examines the use of low-cost PM sensors as ground data sources for estimating surface concentrations from satellite AOD retrievals via two case studies. Specifically, we seek to quantify to what extent, even with the inherent uncertainties of low-cost sensors, their data might still be useful in estimating surface $PM_{2.5}$

from AOD.

First, using a dense network of low-cost monitors in Pittsburgh, Pennsylvania, USA, where a regulatory-grade monitoring network already exists, we assess the utility of low-cost sensors as compared to these traditional instruments. Second, using

low-cost monitors deployed in Rwanda, Malawi, and the Democratic Republic of the Congo, we explore the utility of these low-cost sensors in previously unmonitored areas. We use US State Department data (publicly available from US government websites as well as the OpenAQ Platform at openaq.org) from regulatory monitors at the US Embassies in Kampala, Uganda and Addis Ababa, Ethiopia to supplement our analysis of the relationship between converted satellite AOD data and surface-level PM$_{2.5}$ across SSA. In this work, we focus on high spatial and temporal resolution satellite data, which best aligns with the capacity of low-cost sensors to provide local air quality information in near-real-time. We do not incorporate meteorological or land use information, as such additional information may not be available in sparsely monitored areas. Further, keeping the model as simple as possible avoids over-fitting a more sophisticated model to its calibration data set, which can limit its generalizability. Instead, we use simple linear AOD to surface PM$_{2.5}$ conversion factors to indicate how low-cost sensors alone may provide additional information to inform conversion of AOD to surface PM$_{2.5}$, particularly in data-sparse domains. The techniques presented here are likely to translate to other data sources (e.g. new regulatory-grade monitors, new geostationary satellites) as they become available in the future.

## 2 Methods

### 2.1 Low-cost PM$_{2.5}$ sensor data

Surface PM$_{2.5}$ data were collected with three types of low-cost sensors (MetOne NPM, PurpleAir PA-II, and Alphasense OPC), as described in Table 1. For data collection, all NPM and most PA-II units were paired with RAMP lower-cost monitoring packages. The RAMP (Real-time Affordable Multi-Pollutant) monitor is produced by SENSIT Technologies (Valparaiso, IN; formerly Sensevere), and has internal gas, temperature, and humidity sensors, along with the capability to interface with external PM monitors (newer models also have internal PM sensors). This allows data collected by these PM monitors to be stored and transmitted over cellular networks by the RAMP. The characteristics and operation of the RAMP are described elsewhere (Zimmerman et al., 2018; Malings et al., 2019a). The ARISense node, manufactured by Quant-AQ (Somerville, MA; formerly manufactured by Aerodyne Research), is a lower-cost sensor package that combines internal gas, humidity, temperature, wind, and noise sensors, together with the Alphasense OPC-N2 PM sensor, and provides internet connectivity for data transmission (Cross et al., 2017). Most low-cost PM$_{2.5}$ data are collected via one of these two systems; the exception is a single independently-deployed PA-II unit in Kinshasa, DRC (see Sect. **Error! Reference source not found.**).

Collected data are down-averaged from their device-specific collection frequencies to a common hourly timescale. Erroneous data identified either automatically (e.g. negative concentration values or unrealistically high or low values) or manually (e.g. devices exhibiting abnormal performance characteristics identified during periodic inspections) are removed. To correct for particle hygroscopic growth effects (i.e. the impact of ambient humidity on the PM mass as measured by the low-cost sensors), previously developed calibration methods (Malings et al., 2019b) were implemented for the NPM and PA-II sensors. Briefly, first, a hygroscopic growth factor is computed using the local humidity and temperature as measured by the low-cost monitor itself, along with an average or typical particle composition. Then, a linear correction is applied to the data based on past

collocations with regulatory-grade monitoring instruments. Utilizing these methods, the uncertainties on hourly average $PM_{2.5}$ concentration are about 4 µg/m$^3$ (Malings et al., 2019b). For the Alphasense OPC sensors, raw bin count numbers were integrated to produce a new concentration estimate for $PM_{2.5}$, and a similar relative humidity correction was applied (Di Antonio et al., 2018). An additional correction factor of 1.69 (for workdays) or 1.39 (for non-work-days) was applied to data collected by NPM sensors in Rwanda, based on previous results showing that current calibration methods tended to underestimate $PM_{2.5}$ there (R Subramanian et al., under review). While we seek to use low-cost sensor data that have been calibrated and validated in accordance with best practices, there remain uncertainties associated with these instruments and inaccuracies compared to regulatory-grade instruments. A major goal of this paper is to assess to what extent, even with these uncertainties, low-cost sensor data might still be useful in the context of conversion of AOD to surface $PM_{2.5}$.

## 2.2 Ground-based sampling locations

Surface $PM_{2.5}$ data analyzed in this paper were collected in seven different areas, as listed in Table 2, where approximate locations, number of sites in each area, and durations of data collection are also listed. Maps of these sites are also provided in the supplemental information (Fig. S4-S9). The Pittsburgh area includes sites in the surrounding Allegheny county, although most sites are concentrated within the city. Similarly, the Rwanda area has most sites located in the capital city of Kigali, with one rural monitoring site collocated with the Mount Mugogo Climate Observatory in Musanze. In the Pittsburgh and Rwanda areas, low-cost sensors are connected with RAMP low-cost monitors. In Malawi, data are collected by three ARISense monitors using Alphasense OPC sensors, deployed to three locations in the vicinities of Lilongwe and Mulanje. The two locations in the vicinity of Mulanje are village-center sites, and so may be influenced by nearby combustion activities. In Kinshasa, a single PurpleAir PA-II was deployed independently (i.e. without an associated RAMP unit, as was the case in Pittsburgh) at the US Embassy. Temperature and humidity data were therefore obtained from the internal sensors within the devices themselves, and data connectivity was achieved using the local wireless internet network. At Kampala and Addis Ababa, regulatory-grade monitoring data collected at US Embassies are used to provide ground comparison data for concentration estimates derived from satellite AOD data. Additional information about all of these areas are also provided in the supplemental information (Sect. S1).

## 2.3 Regulatory-grade instrument data

At several locations in the Pittsburgh area, as well as at the US Embassy locations in Kampala and Addis Ababa, hourly-averaged ground-level $PM_{2.5}$ data are also available from regulatory-grade monitoring instruments. In Pittsburgh, these monitors are operated by the Allegheny County Health Department (ACHD). At the US Embassies, these instruments are operated by the US State Department and US EPA and data are made available by these agencies (https://www.airnow.gov/international/us-embassies-and-consulates), as well as by the OpenAQ Platform (openaq.org). In all cases, regulatory-grade monitoring data are collected with Beta Attenuation Monitors (BAMs), a federal equivalent monitoring method, that provide hourly $PM_{2.5}$ concentration measurements for air quality index calculation purposes (Hacker, 2017;

McDonnell, 2017). Nominally, such federal equivalent methods are required to be accurate within 10% of federal reference methods (Watson et al., 1998; US EPA, 2016). Since BAM data have been used to establish the calibration methods for low-cost PM sensor data (Malings et al., 2019b), the data from the BAM instruments are used as provided for uniformity, without any additional corrections being applied.

## 2.4 Satellite data

The satellite data product used in this paper is the MODIS MCD19A2v006 dataset (Lyapustin and Wang, 2018) available through NASA's Earth Data Portal (earthdata.nasa.gov). This dataset consists of AOD information for the 470nm and 550nm wavelengths from the MODIS system, processed using the Multi-angle Implementation of Atmospheric Correction (MAIAC) algorithm, and presented at 1 km pixel resolution for every overpass of either the Aqua or Terra satellites (Lyapustin et al., 2011a, 2011b, 2012, 2018). This represents a Level 2 data product, meaning that it includes geophysical variables derived from raw satellite data, but has not yet been transformed to a new temporal or spatial resolution, as is the case for data derived from multiple satellite passes, e.g. monthly average AOD data. Data from identified cloudy pixels are masked as part of the data product; possible misidentification of cloudy pixels is one source of error in relating surface $PM_{2.5}$ and AOD. As per recommendations in the User Guide for this dataset, only data matching "best quality" quality assurance criteria are used. This dataset was chosen as it represents the highest possible spatial and temporal resolution for AOD, thus providing the most points for comparison with the high spatio-temporal resolution low-cost monitor data.

Satellite AOD data are considered to be collocated in space with data from a ground site when the center of the AOD pixel is within 1 km of the ground site. Data are considered concurrent if the satellite overpass occurs within the hour interval over which ground site data have been averaged to arrive at the hourly-average $PM_{2.5}$ concentration value used. As we compare data from individual satellite passes directly to temporally collocated ground site data, we do not need to consider (as would be essential for long-term averages) the potential impact of the fraction of time where satellite measures are missing (due to cloud cover or other factors). Likewise, we do not consider the biases associated with the fact that satellite passes occur at certain times of day (required when comparing with daily-averaged ground monitoring data) since here we only compare AOD to surface $PM_{2.5}$ during the same hour when the satellite pass occurs.

## 2.5 Conversion Methods for satellite AOD

A linear regression approach is used to establish relationships between satellite AOD and surface-level $PM_{2.5}$. Let $y_{i,t}$ denote the ground-level $PM_{2.5}$ measurement at location $i$ and time $t$, and let $x_{i,t}$ represent the satellite AOD (i.e., a vector combining the AOD at 470nm or 550nm wavelength with a "placeholder" constant of one to allow fitting of affine functions) corresponding to location $i$ and time $t$. For this paper we present results using AOD at 550nm; results for AOD at 470nm are similar and are included in the supplemental information (Sect. S3.2). The total set of ground measurement sites in an area, $S$, is partitioned into two disjoint sub-sets. Subset $S_{in}$ represents the sites used to establish the linear relationship between AOD

and surface PM$_{2.5}$ concentrations. The remainder of sites, in the subset $S_{ap}$, are used for the application, i.e., to serve as an independent set to evaluate the performance of the linear relationship established from the $S_{in}$ sites. Likewise, the time domain

$T$ is partitioned into initialization phase $T_{in}$, during which linear relationships are established, and application phase $T_{ap}$, during which these relationships are applied and evaluated.

Linear relationships are determined as follows. First, satellite AOD data and surface PM$_{2.5}$ monitor data from the $S_{in}$ sites during the $T_{in}$ phase are collected together:

$$X_{in} = \{x_{i,t}\} \quad Y_{in} = \{y_{i,t}\} \quad \forall\ i \in S_{in}\ ,\ t \in T_{in}, \tag{1}$$

A linear relationship is established between these, defined by parameters $\beta_{in}$, using classical least-squares linear regression (e.g., Goldberger, 1980):

$$\beta_{in} = \left(X_{in}^{T} X_{in}\right)^{-1} X_{in}^{T} Y_{in}, \tag{2}$$

The covariance matrix of the parameters, $\Sigma_{\beta_{in}}$, is also obtained:

$$\Sigma_{\beta_{in}} = \frac{(Y_{in} - X_{in}\beta_{in})^{T}(Y_{in} - X_{in}\beta_{in})}{\text{length}(Y_{in}) - \text{length}(\beta_{in})} \left(X_{in}^{T} X_{in}\right)^{-1}, \tag{3}$$

where $\text{length}(\cdot)$ is a function returning the number of elements in the input. During the application phase, the linear relationship can be used to estimate the surface PM$_{2.5}$ concentration at location $i$ and time $t$, $\hat{y}_{i,t,\text{prior}}$, from the satellite AOD data corresponding to that location and time:

$$\hat{y}_{i,t,\text{prior}} = x_{i,t}\, \beta_{in}, \tag{4}$$

The above procedure constitutes an offline or (in Bayesian terminology) prior conversion, i.e., it uses data collected during the

initialization phase to define a single conversion factor that is applied throughout the application phase. An online, dynamic, or (in Bayesian terminology) posterior approach can also be adopted, in which this relationship is modified as additional data are available. This approach has been proposed by Lee et al. (2011) and evaluated by Han et al. (2018), and allows for the potentially time-varying relationship between satellite AOD and surface PM$_{2.5}$ concentration to be accounted for. In the online approach, for a time $t$ during the application phase, a new data set consisting of $Y_{in,t}$ and $X_{in,t}$ is created by combining all data

available from the $S_{in}$ ground sites together with satellite AOD data for that time:

$$X_{in,t} = \{x_{i,t}\} \quad Y_{in,t} = \{y_{i,t}\} \quad \forall\ i \in S_{in}, \tag{5}$$

Based on these new data, a linear relationship is established for that time, as above:

$$\beta_{t} = \left(X_{in,t}^{T} X_{in,t}\right)^{-1} X_{in,t}^{T} Y_{in,t}, \tag{6}$$

This relationship is combined with the prior relationship established during the initialization phase (using a Bayesian approach

and assuming normally-distributed parameter values) to establish a new posterior relationship specific to that time, $\beta_{t,\text{post}}$:

$$\beta_{t,\text{post}} = \beta_{\text{in}} + \Sigma_{\beta_{\text{in}}} \left( \Sigma_{\beta_{\text{in}}} + \eta^2 \text{diag}(\Sigma_{\beta_{\text{in}}}) \right)^{-1} (\beta_t - \beta_{\text{in}}) \approx \frac{1}{1+\eta^2} (\eta^2 \beta_{\text{in}} + \beta_t), \tag{7}$$

where $\text{diag}(\cdot)$ denotes a matrix diagonalization and $\eta$ is a relative error scale parameter, used to define how much "weight" is given to the time-specific relationship parameters $\beta_t$ versus the prior relationship parameters $\beta_{\text{in}}$ in the updating process (with values of $\eta$ near zero placing more weight on the time-specific relationships, while high values of $\eta$ place more weight on the prior). The posterior relationship is then used to estimate surface PM$_{2.5}$ concentrations from the satellite AOD data for that time:

$$\hat{y}_{i,t,\text{post}} = x_{i,t} \, \beta_{t,\text{post}}, \tag{8}$$

Both the offline and online approaches are used in this paper, and their performance is compared (see Sect. 3.1).

This simple linear correction factor method does not explicitly account for vertical distribution profiles, cloud cover, or any other variables that affect the relationship of AOD to surface PM$_{2.5}$. Instead, the aggregate effect of these variables is accounted for implicitly in an empirical relationship. The offline approach uses fixed relationships, which cannot account for time-varying effects such as changes in vertical distribution profiles. The online approach can account for these time-varying effects by assuming their observed impact on the AOD to surface PM$_{2.5}$ relationship at the $S_{\text{in}}$ sites is representative of their short-term impact throughout the region where the corresponding correction factors are applied. Finally, note that all parameters described above can be solved for analytically using the equations presented in this section (i.e. no iterative or approximate solution methods are necessary).

## 2.6 Analyses conducted in this paper

This section provides details of how the various analyses and comparisons to be discussed in Sect. 3 are performed. Additional details are also provided in the supplemental information (Sect. S2.2 to S2.4).

### 2.6.1 Comparison of regulatory and low-cost monitors as ground stations to develop conversion factors for AOD

Here, we seek to compare the performance of AOD conversion to surface PM$_{2.5}$ using either low-cost or regulatory-grade monitors as the ground-level data source for initialization. As only Pittsburgh has networks of both types of sensors in place, we focus our analysis in this area. The surface PM$_{2.5}$ data collected at the five ACHD regulatory monitoring locations are used to assess the performance of the satellite AOD conversion, regardless of how the conversion factors are initialized. First, we use four of five ACHD locations to develop a conversion factor and apply it to the fifth. All ACHD sites are rotated through in this manner, providing a performance metric assessed for AOD conversion applied to each site. Second, we use low-cost sensors for developing the conversion factor; in this case, we select a subset of four locations in Pittsburgh where RAMP low-cost monitors are deployed, so that the number of ground sites used matches the number of ACHD sites used in the first case. These low-cost monitor locations are chosen to provide a similar spatial coverage over Allegheny county as the ACHD sites. Low-cost monitors collocated with ACHD sites were specifically not chosen to allow for a fairer comparison when

performance is assessed against these ACHD site (since, if this were not done, it would be possible to have initialization sites which are collocated with the application sites, which was not possible when the ACHD sites alone were used). In this case, a conversion factor developed using the four low-cost sensor sites is applied at all five ACHD sites, with performance assessed at each site.

Different application cases of the satellite AOD conversion method are also tested. Note that in either case, we use all the collocated ground and satellite data across the entire time period without averaging these data in time. For a "yearly" conversion, data from the entire calendar year are used to develop the conversion factors, while in the "monthly" case, data from the previous month are used to develop conversion factors that are then assessed in the current month (e.g. January data are used to develop conversion factors that are applied in February, then the February data are used to develop conversion

factors that are applied in March, etc.). For the "monthly" case, the median performance across months is presented. Although the "yearly" case would technically require having access to data that have not yet been collected (assuming this method is being applied for data collected in the current year), we use this to represent a case where data from a previous year are used to develop conversions applied in the current year, as we assume that the annual average AOD to surface $PM_{2.5}$ concentration relationship for a given area will not significantly change from one year to the next. In addition, we also assess the relative

performance of the offline (prior) conversion factors, where the same relationship parameters determined during the initialization period are applied to the entire application period, and the online (posterior, dynamic) conversion, where these initial parameters are modified based on the AOD to surface $PM_{2.5}$ relationships specific to each individual satellite pass. The results of this analysis are discussed in Sect. 3.1.

### 2.6.2 Analysis of AOD conversion factor performance versus number of ground sites

A significant advantage of low-cost monitors compared to traditional instruments is that we can deploy tens to hundreds of low-cost sensors for the price of a single regulatory-grade monitor. To assess the potential benefits of this in terms of conversion of satellite AOD data to surface $PM_{2.5}$, we analyze the influence of the number of surface sites used on the performance of the surface $PM_{2.5}$ estimates from AOD conversion. We again examine the Pittsburgh region, vary the number of ground sites used for initialization to generate the AOD conversion factor, and evaluate the performance using the ACHD

regulatory monitoring network as the "ground truth. For the ACHD network, the possible sites are the ACHD sites minus the one site against which performance is assessed (all ACHD sites are rotated through). For the low-cost sensors, the possible sites are all RAMP deployment locations in the area, excluding RAMPs that are collocated with ACHD sites, and performance is assessed against all ACHD sites. Subsets of varying size are randomly selected (10 different random set selections are used in this example); the mean of the performance metric across these 10 randomly selected sets is used as the assessed

performance. In this case, a yearly online conversion factor is used (based on the performance of that method as described in Sect. 3.1). The results of this analysis are discussed in Sect. 3.2.

### 2.6.3 Comparison of converted AOD and nearest ground monitors as proxies for surface PM$_{2.5}$

Here, we seek to assess the benefits of combining satellite AOD and ground-based sensor data, as compared to using ground-based sensor data alone. For this assessment, we compare estimates of surface PM$_{2.5}$ derived from satellite AOD data, using the methods presented previously in this paper, with estimates based on the surface PM$_{2.5}$ measurements alone, which we denote as "nearest monitor" estimates. For this estimation, we make use of a locally constant or naïve interpolation, in which the surface PM$_{2.5}$ estimate for a given time and location is the same as the measurement of the nearest available ground monitor (i.e., one of the ground monitors used for establishing conversion factors for the satellite AOD data) at that time:

$$\hat{y}_{i,t,\text{nearest}} = y_{j,t} \text{ s.t. } j = \text{argmin}_{k \in S_{\text{cal}}} \text{dist}(i,k), \tag{9}$$

where $\text{dist}(i,k)$ indicates the distance between locations $i$ and $k$, and $\text{argmin}$ denotes the input that minimizes this objective. In this case, low-cost sensor data are used to represent the "ground truth" against which performance is assessed; this is done so that a comparable analysis can be made in Pittsburgh and Rwanda, since no regulatory-grade instruments were present in the latter area. Prior conversion factors are developed for the entire time period and are updated to posterior factors with time-specific data for their application. All but one low-cost sensor sites in a given area are used for development of these factors, with application and assessment on the final site. These sites are then cycled through, to provide performance metrics across all sites. To allow for comparability between the nearest monitor approach and surface PM$_{2.5}$ estimation from satellite AOD, we make use of the same set of ground sites for both, i.e., for each site, data from the closest available sites are used as inputs to the nearest monitor method, and all sites are cycled through in this manner, providing performance metrics for each site as above. The results of this analysis are discussed in Sect. 3.3 (for Pittsburgh) and 3.4 (for Rwanda).

### 2.6.4 Analysis of inter-seasonal generalization of AOD conversion factors

Changing seasons can affect the relationship between satellite AOD and surface PM$_{2.5}$ due to changes in confounding factors like surface reflectance, aerosol vertical profiles, and particle composition. Here, we assess the utility of developing seasonal AOD conversion factors for Pittsburgh and Rwanda. For this assessment, conversions are developed and applied in specific seasons (information on these seasons are presented in the supplemental information, Table S1 and Fig. S1). For Pittsburgh, these approximately correspond to a winter, spring, summer, and fall season, while in Rwanda, these represent alternating wet and dry seasons. For Pittsburgh, the major differences between seasons are related to temperature, with humidity varying to a lesser degree. In Rwanda, temperatures are relatively stable year-round, with seasons mainly differentiated by humidity changes (although the second dry season appears to have been unusually wet, comparable to the previous wet season).

RAMP data are used to represent "ground truth" concentrations for both areas. An offline or "prior" approach is used here, i.e., calibrations are not modified based on data collected within the application period, in order to investigate the effect of generalizing a calibration developed in one season to a different season. Metrics are assessed for each individual site in each area, with all other sites being used to establish AOD conversion factors as in the previous section. The results of this analysis are discussed in Sect. 3.5.

### 2.6.5 Analysis of inter-regional generalization of AOD conversion factors

Finally, given the lack of ground-based monitoring in many parts of SSA, we assess whether a conversion factor developed in one city of this region can be generalized to other cities or locations across SSA. Here, a single AOD conversion factor is developed using one site in Kigali, Rwanda and this factor is applied without modification to other sites across SSA. These include a second site in Kigali, a site in Musanze in rural Rwanda, a site in Kinshasa (DR Congo), and three sites in Malawi (one near the urban area of Lilongwe and two other sites in more rural areas to the south, near Mulanje) where low-cost sensor

systems are deployed. There are also two sites (Kampala, Uganda and Addis Ababa, Ethiopia) where hourly-resolution long-term regulatory-grade monitoring data are available; data from these sites are included for comparative purposes. An offline approach is used here, with a single factor being initialized over the entire study period. The results of this analysis are discussed in Sect. 3.6.

### 3 Results

In this section, we apply the proposed method for satellite AOD to surface $PM_{2.5}$ concentration conversion in several use cases. In Sect. 3.1, 3.2, and 3.3, we assess the performance in Pittsburgh, comparing the use of regulatory-grade monitors and low-cost monitors as ground sites for establishing conversion factors. In Sect. 3.4 and 3.5, we extend the comparison to Rwanda, examining the impact of using the relatively sparser low-cost sensor network there, and examining seasonal variations in the conversions. Finally, in Sect. 3.6, we examine the generalization of Rwanda-based conversion factors to other locations across

SSA. Assessment metrics used in this section, including correlation (r), coefficient of variation of the mean absolute error (CvMAE), and mean-normalized bias (MNB) are described in the supplemental information (Sect. S2.1).

### 3.1 Comparing the use of regulatory and low-cost monitors as ground stations to develop conversion factors for AOD

We first evaluate the utility of low-cost sensors as substitutes for regulatory-grade monitors when developing factors to convert satellite AOD data to surface $PM_{2.5}$ estimates, using the Pittsburgh area as our case study. Results for all eight combinations

of ground initialization site monitor type ("ACHD" v. "RAMP"), initialization period length ("yearly" vs. "monthly"), and application mode ("prior" vs. "post.") are presented in Fig. 2. Overall, these results indicate relatively weak relationships between satellite AOD and surface $PM_{2.5}$ for Pittsburgh, regardless of the method used. Correlations are weak (r < 0.5) and mean absolute errors are on the order of half to three-quarters the concentration values (annual average concentrations are about 10 $\mu g/m^3$ across most of Pittsburgh). Biases are low on average but can vary across locations. In comparing the different

application modes, the "posterior" method provides better performance in terms of correlation than the "prior" method. This suggests that variability in AOD to surface $PM_{2.5}$ relationships between satellite passes (e.g., due to differences in the vertical profile of $PM_{2.5}$ over the area, and/or to differences between "point" measurements of the ground monitors and "area" AOD) is better captured by updating prior relationships with new information from each new satellite pass. In terms of other performance metrics, there is little difference between these application modes, with slight improvements observed in the

"posterior" method for the RAMP data, but slight decreases for the ACHD data. Comparing the use of annual to monthly initializations, performance metrics are slightly worse in the monthly case, indicating that the additional initialization data used in the yearly case generally leads to a more robust conversion. It should be noted, however, that these conclusions may be specific to relatively low $PM_{2.5}$ concentrations as found in Pittsburgh.

In all cases, performances using low-cost sensor data are comparable to that of the same conversion approaches utilizing the regulatory-grade instruments. Note that the low-cost monitors used here have been carefully corrected by collocation with regulatory-grade monitors (Malings et al., 2019b) which accounts for known artefacts with low-cost sensors. Thus, there is no evidence from this analysis of any inherent disadvantage to the use of carefully corrected low-cost sensors to provide ground data as compared to more traditional instruments. Rather, based on these results, any additional uncertainty due to data quality differences between low-cost sensors and regulatory-grade instruments are seen to be negligible compared to the difficulties associated with relating satellite AOD to surface-level $PM_{2.5}$, and therefore have had no systematic impact on the performance of the assessed linear conversion method, at least for this study area.

### 3.2 How many ground stations are needed to improve surface $PM_{2.5}$ estimates from AOD retrievals?

Fig. 3 sh For small numbers of ground sites, results for the ACHD network and the low-cost sensor network are similar in terms of mean performance across different randomly selected subsets of the network, with slightly better performance using the RAMP network sites. This may be related to the smaller number of possible combinations of ACHD sites to be randomly selected compared to the RAMP sites; with more RAMP sites to choose from, the likelihood of selecting more generally representative (rather than more source-impacted) sites is higher, whereas with the ACHD network there is a high likelihood of choosing a heavily source-impacted site (especially since several ACHD locations are specifically chosen to monitor such local sources; see supplemental information Fig. S4). The limited number of ACHD sites prevents this analysis from being expanded to larger numbers of locations; at four chosen locations, there is only one possible combination to be selected, and so the spread in performance collapses to match the mean. With the low-cost sensor network, as more ground sites are included, mean CvMAE decreases until about 10 sites are chosen, but afterwards remains relatively constant as more sites are included. Performance variability decreases as more site are added, indicating that by adding additional ground sites, even sites positioned at random throughout the domain, the conversion relationship becomes increasingly robust. While for a single ground monitor, worst-case CvMAE is on the order of 1.5 to 2, with 10 or more monitors, worst-case performance is improved below 0.6, a more than two-fold improvement in worst-case performance. Overall, this demonstrates the potential benefits of dense low-cost sensor networks for conversion of satellite AOD data, even over a limited spatial domain (covering about 600 $km^2$). Furthermore, it shows that even with quasi-random placement of the ground sites, such as might be achieved by citizens making personal decisions to deploy low-cost monitors on their own properties, increasingly robust conversion results can be achieved as more sensors are included, although these benefits diminish beyond (at least in the case of Pittsburgh) about 1 monitor per 60 $km^2$.

### 3.3 Comparison of AOD-based surface PM$_{2.5}$ to measurements from a dense ground network

Performance of both the nearest monitor method and the satellite AOD conversion method are assessed for Pittsburgh in Fig. 4. It should be noted that all available ground sites have been used for conversion factor initialization in this section, versus a limited subset of these in Sect. 3.1, leading to improved performance of this method following the trend noted in Sect. 3.2. In Pittsburgh, we see reduced performance (lower correlation, larger CvMAE) when using converted satellite data as compared to nearest monitor data. This is likely a result of the quite dense network of low-cost sensors present in Pittsburgh, where the median distance between sensors in the network is about 1 km. With this dense network, there is a good chance that the nearest ground monitor will be quite close to the location at which concentrations are to be estimated, and the resulting "nearest monitor" estimate is therefore likely to be quite good, as PM concentrations tend to be homogenous at this spatial scale in Pittsburgh (Li et al., 2019). When PM$_{2.5}$ is instead estimated from satellite data using a simple linear relationship, spatial and temporal variability in surface PM$_{2.5}$ to AOD relationships can confound the assessment. This is especially important considering the relatively low levels of surface PM$_{2.5}$ concentration and AOD in and above Pittsburgh, meaning that any introduced noise will be relatively large in proportion to the signal being assessed. These results indicate that dense ground-based monitoring (if available) will likely outperform AOD-derived surface PM$_{2.5}$ at least for the simple conversion method used here.

### 3.4 The utility of AOD-based surface PM$_{2.5}$ in regions with a sparse ground monitoring network

Performance of the nearest monitor method and the satellite AOD conversion method are assessed for Rwanda in Fig. 5, in a similar manner as was done for Pittsburgh in Fig. 4. In Rwanda, we see an improvement across all metrics (higher and more consistent correlation, smaller and more consistent CvMAE, and less spread in the bias) as satellite data are combined with surface PM$_{2.5}$ monitor data. Median CvMAE is reduced from about 0.5 to 0.3, a 40% improvement. Because of the relative sparsity of the low-cost monitor network in Rwanda (4 measurement sites, not all of which were simultaneously operational) compared to that in Pittsburgh, the assumption of spatial homogeneity of concentrations between monitoring sites is less valid, and so the inclusion of satellite data is beneficial in resolving these spatial differences. Furthermore, the relatively high levels of PM$_{2.5}$ concentration in Rwanda (average of about 40 µg/m$^3$ over the study period) allows for a higher signal-to-noise ratio relative to Pittsburgh. Together, these results indicate the high utility of low-cost sensors, used in conjunction with satellite data, when these are deployed even in relatively sparse networks to previously unmonitored areas with high surface PM$_{2.5}$ concentrations.

This point is further explored in Fig. 6, which compares the correlations between ground measurements in Pittsburgh and Rwanda with the AOD-to-surface-PM correlations in these areas. In Pittsburgh, the high density of available monitors leads to relatively high inter-site correlations, above the typical range of the AOD-to-surface-PM correlations. It is therefore difficult to extract meaningful patterns from the AOD information that would not also be present in available surface-level measurements, suggesting that AOD data provide little additional value in this densely monitored area (at least in terms of

what can be derived without including additional information sources like atmospheric modelling and land use characteristics).

Meanwhile, in sparsely monitored Rwanda, inter-site correlations are lower, overlapping the typical range of AOD-to-surface-PM correlations. This means that AOD data can still provide useful information for spatial heterogeneities in this region.

## 3.5 Seasonal effects on satellite AOD conversion to surface PM$_{2.5}$

Fig. 7 presents the median performance metrics across all sites in either Pittsburgh or Rwanda for each combination of initialization and application season. For Pittsburgh, spring conversion factors seem to generalize best when applied to other

seasons, with the lowest biases and highest precisions. Low correlations are observed in the summer and winter regardless of initialization period, and clear seasonality is observed with summer initializations being biased high in winter and winter initializations being biased low in summer.

In Rwanda, an alternating pattern is revealed, with wet season conversion factors applying well to other wet seasons, and dry season conversion factors applying to other dry seasons. Many factors could contribute to this pattern, including changes in

humidity and the resulting impact on extinction, as well as seasonal burning patterns affecting particle sizes and compositions. Conversion factors appear to generalize better between wet seasons than between dry seasons. Correlations are highest during the first dry season (DS1), regardless of whether the conversion factor is developed during this season or during the surrounding wet seasons; this was also the driest season and the season with the highest PM$_{2.5}$ concentrations. Applications of conversion factors developed in other seasons to DS1 underestimate PM$_{2.5}$ in this season, especially applications of factors developed

during the wet seasons (when PM$_{2.5}$ levels were much lower). This indicates that there is seasonality to PM$_{2.5}$ concentrations that is not being reflected in the AOD data, and requires local monitoring to identify. Overall, these results indicate that conversion factors should be developed or updated at least on a seasonal basis, especially in Rwanda; a conversion factor developed during a limited monitoring campaign occurring in one specific season may fail to generalize well to other seasons.

## 3.6 Regional generalization of AOD conversion factors developed in Rwanda

Results of the analysis discussed in Sect. 2.6.5 are presented in Fig. 8. Correlation is relatively low across most application areas, with a weak trend of decreasing correlation as distance from the initialization site increases (the exception to this is found at the rural Mugogo site). Best performance in terms of CvMAE and normalized bias is found in Kigali, Kampala, and Kinshasa; these urban zones are likely most similar to the initialization site in terms of land use and resulting source mix. Relatively best performance is found at the spatially closest Kigali site. The Kampala site, with data collected via a traditional

monitoring instrument, shows similar results as obtained at these other urban sites with low-cost monitors. The other, more rural locations show poorer performance regardless of distance from the initialization site. However, the Addis Ababa site also shows much poorer performance, despite also being an urban area, although the Embassy is located on the outskirts of the city. This may be due to climate differences between Addis Ababa and the other cities considered, as well as differences in particle composition and size distributions, especially higher contribution to AOD from coarse (larger than PM$_{2.5}$) Saharan dust (De

Longueville et al., 2010) that would not be accounted for in the Kigali-based AOD conversion factor.

These results indicate that, while conversion factors may generalize to sites with similar land use and climate characteristics, physical distance alone is not as significant in determining AOD-PM relationship generalizability. Also, the overall low correlation values indicate the importance of local data, as spatial heterogeneity in satellite AOD to surface $PM_{2.5}$ relationships can be a concern even for nearby sites. Finally, it should be noted that a single annual conversion factor, as is assessed here, could fail to take into account seasonal variabilities (Sect. 3.5) and so can correlate poorly with surface $PM_{2.5}$ even in or near the area where it is developed (as seen for the Kigali site here). A conversion factor that varies on at least a seasonal basis is therefore preferred; however, determining how to generalize such a time-varying conversion factor to other regions where seasonal definitions and characteristics can be quite different is a challenging problem. Overall, it does not appear from this analysis that AOD to surface $PM_{2.5}$ conversion factors can be broadly generalized across global regions with consistent results. Therefore, continuous localized monitoring, such as might be facilitated with local low-cost monitor networks, seems to be the most robust way to establish applicable AOD to surface $PM_{2.5}$ conversion factors.

## 4 Discussion

We have examined the feasibility of using low-cost sensors as a data source in developing relationships between surface $PM_{2.5}$ concentrations and satellite AOD. In a case study in Pittsburgh, there was no decrease in performance associated with the use of low-cost sensors for this purpose rather than more traditional regulatory-grade monitors, although performance was rather poor in both cases. The higher density ground networks possible with low-cost sensors did provide benefits in terms of more robust conversion factors compared to the more sparsely deployed traditional monitoring network. However, it was found that for Pittsburgh, with a relatively dense low-cost sensor network (median inter-site distance of about 1 km) and low $PM_{2.5}$ concentrations, use of the nearest ground measurement sites outperformed the use of satellite AOD data to estimate surface $PM_{2.5}$ using linear conversions. Partly, this could be because AOD is rather low over this area (average of about 0.2) leading to lower signal-to-noise ratios that reduce AOD to surface PM correlation. Conversely, in Rwanda, a relatively sparse low-cost sensor network combined with satellite data in an environment with higher and more variable $PM_{2.5}$ concentrations provided better estimates of surface $PM_{2.5}$ concentrations than was available using only the nearest surface monitor alone. This result is highly relevant to SSA, as sparse local monitoring and high average $PM_{2.5}$ concentrations (as measured by the few available ground-based monitors) are common features. Differences in seasonal characteristics (especially at the Rwanda locations) show the added value of season-specific conversion factors (which are facilitated by continuous local monitoring), while differences in characteristics between areas, especially urban and rural locations with highly variable particle types, limit the generalizability of conversion factors across regions (again emphasizing the importance of local monitoring).

The results presented here continue to highlight the need for ground-based $PM_{2.5}$ monitoring in previously unmonitored areas such as SSA, especially in light of the benefits observed in Rwanda from having even a sparse ground monitoring network combined with satellite data on local spatial heterogeneity. Efforts to expand ground-based monitoring should make use of traditional regulatory-grade instruments wherever possible, supplemented with low-cost monitors to increase network density

and expand spatial coverage. Findings in Pittsburgh indicate that denser monitoring networks, such as those made possible by low-cost sensors, improve accuracy and robustness of surface $PM_{2.5}$ estimates from satellites. Verification that the same trend

will hold in other regions, especially in SSA, requires further dense deployments of low-cost sensors, and is the subject of ongoing work.

It should be noted that the results of this paper pertain to local and instantaneous relationships, using the highest spatial and temporal resolution of satellite data currently available. Results may differ for spatially or temporally aggregated satellite and ground site data. In fact, such spatial and temporal aggregation is likely to reduce the impact of noise (but not bias) both from

low-cost instruments and from satellite retrievals. However, such aggregate information does not take full advantage of the potential inherent in low-cost sensors to provide near-real-time information on local air pollution. On a related point, satellite data (at least, for most of the world using current platforms) cannot provide diurnal concentration profiles, instead presenting a "snapshot" of concentrations for a wide spatial domain but only for a specific time of day. Ground-based continuous monitoring, even with low-cost sensors, will still be essential, at least until new geostationary platforms with truly global

coverage are available (Judd et al., 2018; She et al., 2020). Such satellites are planned for coverage of North America (the TEMPO satellite mission), Europe (Sentinel 4), and East Asia (GEMS); unfortunately, there are no current plans for coverage of Africa by similar satellites.

Further technical and research developments in this area have enormous promise for improving our understanding of local air quality worldwide. A functioning system for converting satellite to ground-level air pollution data, relying on a group of

"trusted" ground data sources, could potentially be a valuable resource for assessing and correcting low-cost sensor data, allowing for in-field recalibration of drifting instruments, and better identification of malfunctioning sensors. Low-cost systems combining PM mass measurement and ground-up AOD data can help to establish AOD to surface PM relationships at finer spatio-temporal resolution (Ford et al., 2019). Open questions related to this research area include finding appropriate timescales over which conversion factors can be considered constant within regions as well as continuing to examine the

question of conversion factor generalizability between regions separated by spatial distances and across different climates and land use characteristics. More sophisticated conversion methods incorporating meteorological and land use information and outputs of chemical transport models can also be considered, albeit with the recognition that some of these inputs may not yet be readily available or well validated for SSA.

**Code and data availability**

Data related to the results and figures presented in this paper are available online at https://doi.org/10.5281/zenodo.3858063. Codes related to the analysis of data and generation of figures are also provided at the same site.

**Author contribution**

Conceptualization: RS, MB; Funding acquisition: RS, AP, MB; Methodology: CM, DW, AG, AB; Resources, DW, AG, AB; Software: CM, AH; Supervision: RS, AP, MB; Writing – original draft: CM; Writing – review & editing: CM, DW, AP, AG, AB, MB, RS.

**Competing Interests**

The authors declare that they have no conflict of interest.

**Acknowledgements**

This work benefited from State assistance managed by the National Research Agency under the "Programme d'Investissements d'Avenir" under the reference "ANR-18-MPGA-0011" ("Make our planet great again" initiative). Measurements in Pittsburgh were funded by the Environmental Protection Agency (Assistance Agreement Nos. RD83587301 and 83628601) and the Heinz Endowment Fund (Grants E2375 and E3145). Measurements in Rwanda were supported by the College of Engineering, the Department of Engineering and Public Policy, and the Department of Mechanical Engineering at Carnegie Mellon University via discretionary funding support for Paulina Jaramillo and Allen Robinson. AG and AB acknowledge support from NSF Award CNH 1923568 to establish measurement sites in Malawi. The authors would like to thank Jimmy Gasore, Valérien Baharane, Abdou Safari Kagabo, Eric Lipsky, Sriniwasa P.N. Kumar, Provat Saha, Yuge Shi, Naomi Zimmerman, Rebecca Tanzer, and S. Rose Eilenberg for assistance with instrument setup and operation. Finally, the authors would like to thank Juan Cuesta and Jiayu Li for discussions and advice related to satellite data usage.

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

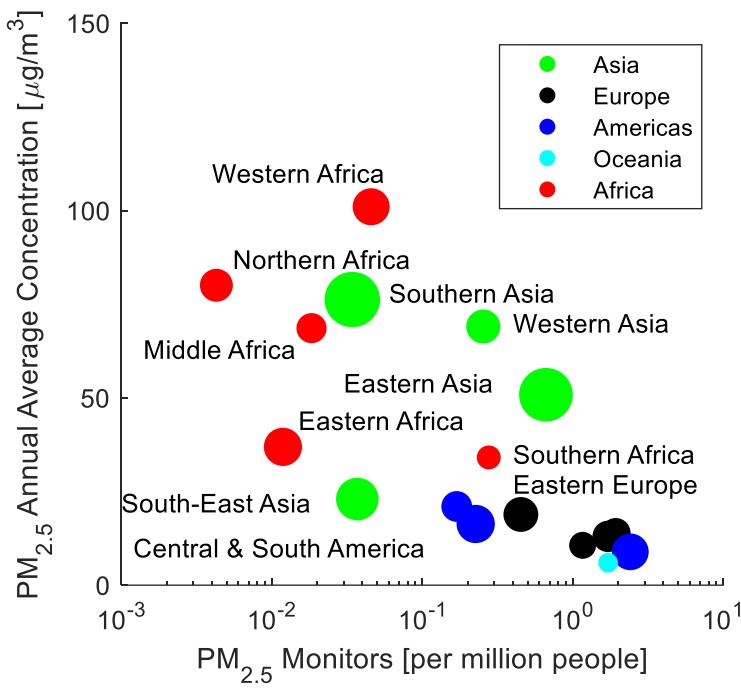

**Figure 1: Estimated annual average PM₂.₅ concentration versus density of regulatory-grade monitoring stations across several global regions. Colors correspond to continents, and sizes roughly correspond to total regional population. This graphic is based on information available from the Global Health Observatory (WHO, 2017).**

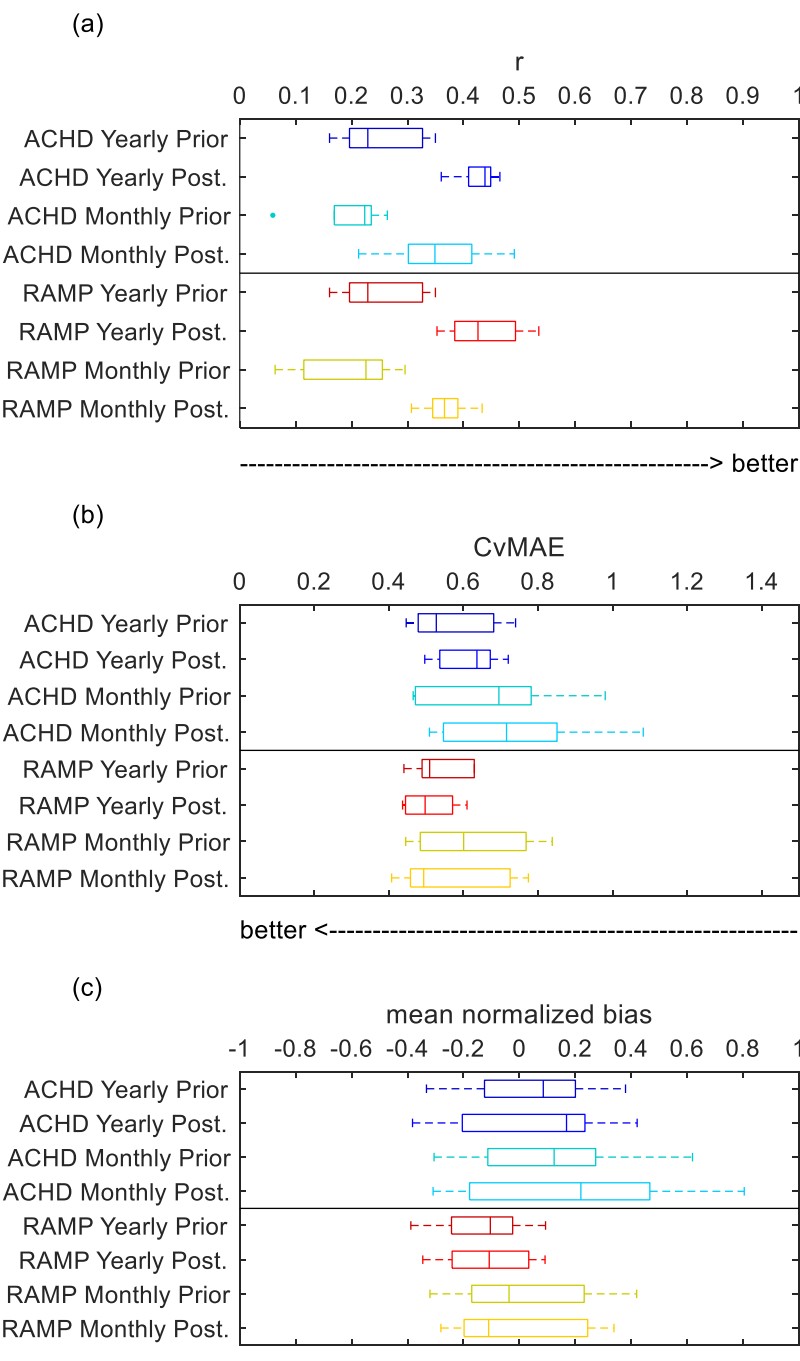

**Figure 2: Comparison of performance metrics (a: correlation, b: CvMAE, and c: MNB) for surface PM₂.₅ estimated from satellite AOD data in the Pittsburgh area. Performance is assessed at the ACHD regulatory-grade monitoring sites. Ground sites used for factor development are either four of the ACHD monitors (ACHD) or four low-cost sensors associated with RAMP monitors (RAMP). Conversion factors are established either on a Yearly or Monthly basis. Finally, either an offline (Prior) or online (Post.) approach is used.**

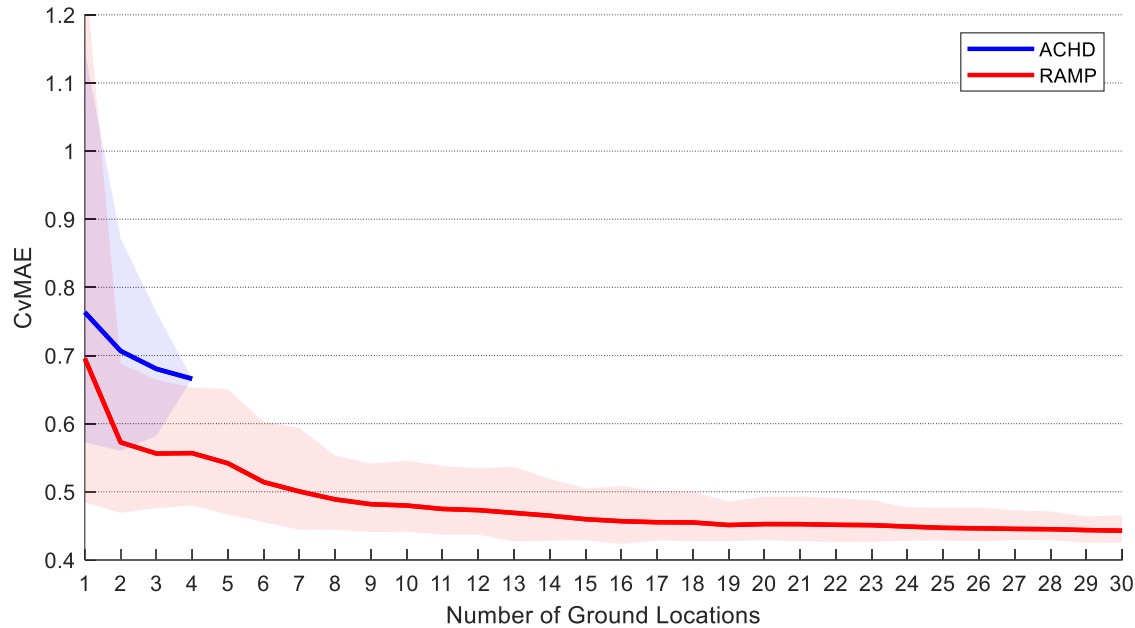

**Figure 3: Performance (assessed in terms of CvMAE) for surface PM$_{2.5}$ estimated from satellite AOD data in the Pittsburgh area, plotted as a function of the number of ground sites used. Performance is assessed against the ACHD regulatory-grade monitors. Solid lines indicate mean performance across sites using either ACHD or low-cost sensor (RAMP) sites to establish conversion factors. Shaded regions indicate the range of variability across application sites.**

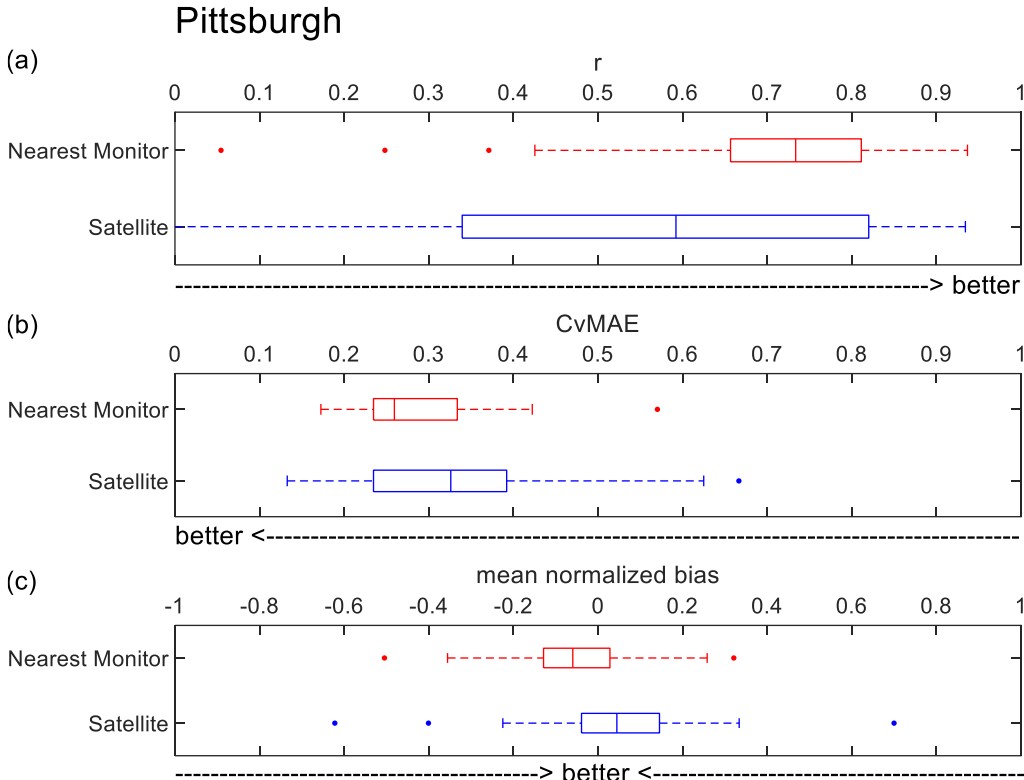

**Figure 4: Comparison of performance metrics (a: correlation, b: CvMAE, c: MNB) for either surface PM$_{2.5}$ estimated from satellite AOD data (Satellite) or from the nearest ground-level PM$_{2.5}$ monitor (Nearest Monitor) in the Pittsburgh area. Note that these performance metrics are not directly comparable to those presented in Fig. 2, as in this case a larger number of ground initialization sites (9 to 45, depending on the number of active sites in Pittsburgh at any particular time) are considered. Further, performance is now being assessed against the RAMP rather than the ACHD network (i.e., performance is assessed at the held-out active RAMP site); this is done to allow for comparability with the results from Rwanda, presented in Fig. 5, where only RAMP data are available.**


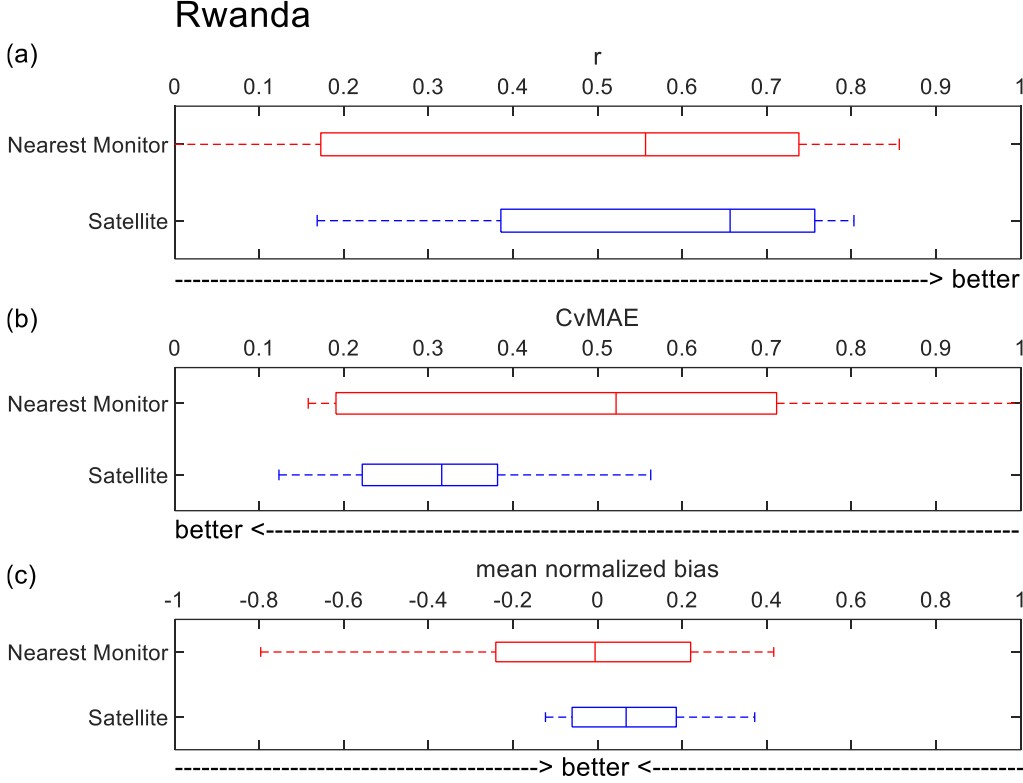

**Figure 5: Comparison of performance metrics (a: correlation, b: CvMAE, c: MNB) for either surface PM$_{2.5}$ estimated from satellite AOD data (Satellite) or from the nearest ground-level PM$_{2.5}$ monitor (Nearest Monitor) in the Rwanda area.**


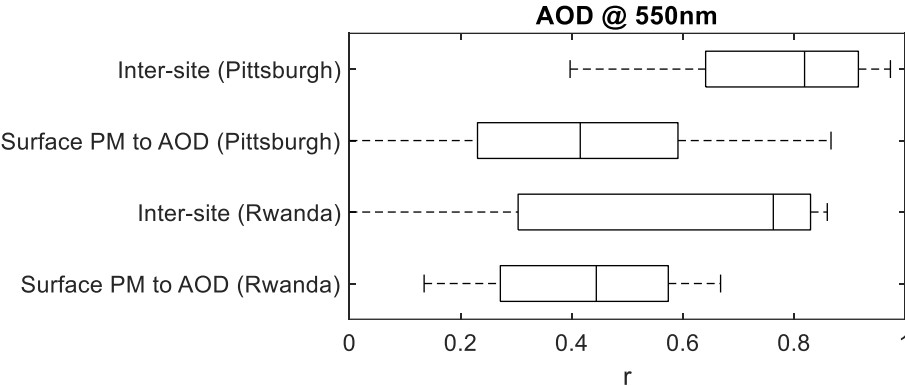

**Figure 6: Comparison of inter-site correlations versus AOD-to-surface-PM$_{2.5}$ correlations in Pittsburgh and Rwanda**


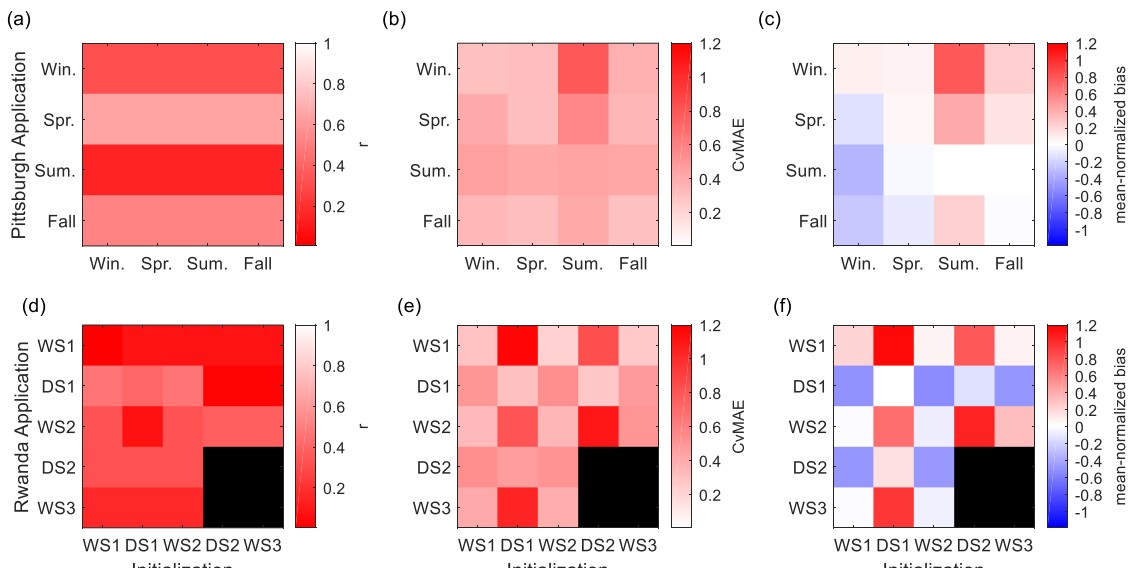

**Figure 7: Comparison of seasonal performance metrics (a, d: correlation; b, e: CvMAE; c, f: MNB) for surface PM2.5 estimated from satellite AOD data across different seasons in the Pittsburgh (a, b, c) and Rwanda (d, e, f) areas. The horizontal axis differentiates the seasons during which initialization was performed, while the vertical axis denotes the seasons when the conversion was applied. Note that, in Rwanda, only one sensor was operational during Dry Season 2 (DS2) and Wet Season 3 (WS3), and so application of these conversions to an independent site was impossible; therefore, performance metrics are blacked out. In each figure diagonal (from top left to bottom right) elements correspond to the same season. Values are also listed in the supplemental information, Table S8.**


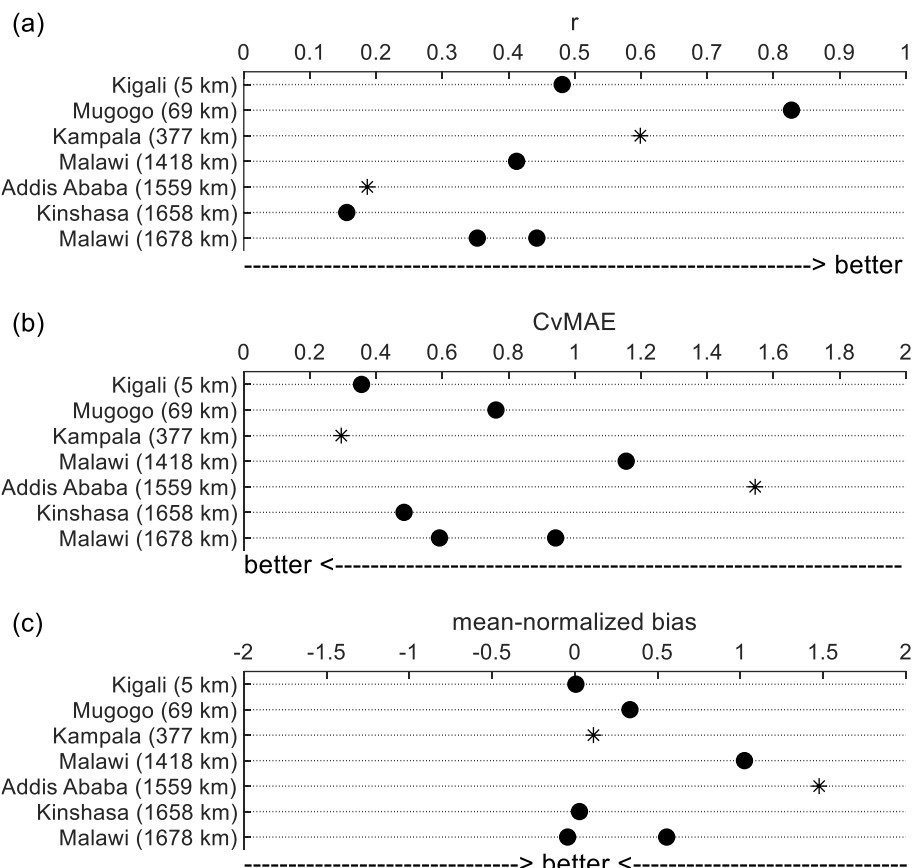

**Figure 8: Comparison of performance metrics (a: correlation, b: CvMAE, c: MNB) for surface PM$_{2.5}$ estimated from satellite AOD data across multiple sites in SSA. The conversion factor is developed at a central site in Kigali, Rwanda; the distances of each testing site to this central site are given. Performances are assessed for all data collected at the given sites, using the prior conversion factor only. Note that performance in Kampala and Addis Ababa is assessed using traditional reference monitors (indicated by ∗), while performance at the other sites reflects low-cost sensor data (indicated by ●).**



**Table 1: Summary information for low-cost sensor systems utilized for this paper.**

| Manufacturer | MetOne | PurpleAir | Alphasense |
|---|---|---|---|
| Product | Neighborhood Particulate Monitor | PurpleAir II | OPC-N2 |
| Abbreviation | NPM | PA-II | OPC |
| Measurement Method | forward light scattering laser | laser particle sensor | optical particle counting |
| Other Features | Includes $PM_{2.5}$ cyclone and inlet heater. Provides estimates of $PM_{2.5}$ mass concentrations using calibrations that are user-modifiable. Interfaced with RAMP low-cost monitor nodes. | Includes a pair of Plantower PMS 5003 units, along with temperature and humidity sensors. Provides estimates of $PM_1$, $PM_{2.5}$, and $PM_{10}$ mass concentrations via proprietary calibrations. Interfaced with RAMP low-cost monitor nodes. | Detects particles in the 0.38 to 17 µm range, converts particle counts to $PM_1$, $PM_{2.5}$, and $PM_{10}$ mass concentrations via proprietary calibrations. Integrated with ARISense low-cost monitor nodes. |
| Unit Cost (approx.) | $2000 | $250 | $350 (not including housing) |
| Performance Notes | Moderate correlation to regulatory-grade instruments in laboratory and field testing. Requires cleaning and re-calibration between deployments. | High correlation to regulatory-grade instruments, except at high humidity. Individual Plantower sensor malfunctions detectable via comparison between the two internal units. | Moderate correlation to regulatory-grade instruments in field conditions. |
| References | (AQ-SPEC, 2015; Malings et al., 2019b) | (AQ-SPEC, 2017; Malings et al., 2019b) | (AQ-SPEC, 2016; Crilley et al., 2018) |

**Table 2: Summary information for the ground sites presented in this paper.**

| Area Name | Pittsburgh | Rwanda | Malawi | Kinshasa | Kampala | Addis Ababa |
|---|---|---|---|---|---|---|
| Country | United States of America | Rwanda | Malawi | Democratic Republic of the Congo | Uganda | Ethiopia |
| Location (Approx.) | Between 40.1ºN, 80.5ºW and 40.8ºN, 79.7ºW | Between 2.2ºS, 29.4ºE and 1.4ºS, 30.5ºE | Between 16.2ºS, 33.6ºE and 14.0ºS, 35.7ºE | 4.3ºS, 15.3ºE | 0.3ºN, 32.6ºE | 9.0ºN, 38.8ºE |
| Start | Jan. 1, 2018 | April 1, 2017 | June 25, 2017 | Mar. 20, 2018 | Jan. 1, 2019 | Jan. 1, 2019 |
| End | Dec. 31, 2018 | May 27, 2018 | July 30, 2018 | Oct. 31, 2019 | Dec. 31, 2019 | Dec. 31, 2019 |
| **Low-Cost Sensors** | | | | | | |
| Total Sites | 62 | 4 | 3 | 1 | | |
| Simultaneously Active Sites | 10 to 46 | 1 to 3 | 1 to 3 | 1 | | |
| Sensor Type | NPM, PA-II | NPM | OPC | PA-II | | |
| **Regulatory-Grade Monitors** | | | | | | |
| Total Sites | 5 | | | | 1 | 1 |
| Type | BAM | | | | BAM | BAM |
