# Peer review of "Application of Low-Cost Fine Particulate Mass Monitors to Convert Satellite Aerosol Optical Depth to Surface Concentrations in North America and Africa"

_Atmospheric Measurement Techniques, 2020_

## Referee Comment (RC1) · Anonymous Referee #1 · 18 Mar 2020

The goal of this paper is to assess the conversion of satellite AOD values (not measurements) to surface PM2.5 concentrations by using low cost sensors and regulation grade instruments. Two regions are selected for study – Pittsburg, Pennsylvania and Rwanda, Africa. Most of the discussion is focused on PA. The paper focuses on linear relationship between AOD and PM2.5

The paper tries numerous avenues to assess the suitability of low cost sensors for surface PM2.5 estimations including space-time constraints on regulation versus low cost sensors and various statistical measures.

Most of my comments are for PA since the paper focuses on this region.

The study region is very small 0.7 degrees by 0.7 degrees. The paper definitely needs a map of some sort showing the location of the regulation grade monitors and the location of the low cost sensors since I have no idea how close are far away these sensors are!

The paper never discusses as to how space-time collocation was done for the ground versus satellite data. The results vary depending upon the width of the time and space windows. The paper also does not provide the slope/intercept values for these linear correlations.

The range of annual values in PA was low and the satellite data and the low cost sensors have larger uncertainties in this range and therefore the results may not be robust. Given this backdrop I am not sure how meaningful the PA results are. This is probably the main reason that the correlations are low – Page 11 (Line 325+).

Not sure about the usefulness of an offline approach where only a single conversion factor is used. Why report these values when we know that this is not relevant?

Page 11, Line 319. What is the cloud cover for each site and how does it affect annual average AOD? Given some of the issues mentioned above I am not sure that page 12 (line 24-244) conclusion is acceptable. Also given that the linear correlation has so many problems, using satellite data and ground monitors to assess the linear relationship is fraught with uncertainties.

In summary, I believe that low cost sensors play an important role for PM2.5 research but unless calibration issues and comparisons with ground monitors of regulation grade are made carefully as a function of space, time, meteorology we cannot be sure how useful the data can be for quantitative monitoring, assessment, and research (e.g. epidemiology). It is also not fair to state that (Page 16, line 482) that using the nearest monitor is better than using satellite data because none of the meteorological factors

have been taken into account for estimating PM25 from satellite data.

Minor comments Wang and Christopher, 2003 – Not Wang, 2003 Some of the references are outdated. E.g. Zhang et al 2009 for correlation coefficients. Page 3 : What spatial/temporal scales did Murray et al used

Page 3 : Not all studies find 'anti-correlation' in India. Page 3: Last sentence needs a reference Page 3: The cloud cover problems needs to be addressed and referenced. Christopher & Gupta (2010) Satellite Remote Sensing of Particulate Matter Air Quality: The Cloud-Cover Problem, Journal of the Air & Waste Management Association, 60:5, 596-602, DOI: 10.3155/1047-3289.60.5.596 Page 4: Errors cannot average out and it depends on the range of PM2.5 values and a host of other factors. Section 2.1.1 to 2.1.3 belongs in a Table rather than a few sentences of text Page 5: Line 1 : Here not hare Page 7 says 'as summarized in 2.1.4' but 2.1.4 does not describe calibration in any detail. Erroneous data screening for negative values is easy but doing this manually for the entire low cost network is not possible. Page 6: Line 180-183 says the data are scaled for workdays and non work days. This type of scaling may work for this study but how about other regions? Page 8: The satellite data needs some description with a proper journal reference. Briefly, how was AOD retrieved, what are the uncertainties, how much cloud cover for the analysis, what quality flags were used, etc.

---

## Referee Comment (RC2) · Anonymous Referee #2 · 24 Mar 2020

The paper "Application of Low-Cost Fine Particulate Mass Monitors to Convert Satellite Aerosol Optical Depth Measurements to Surface Concentrations in North America and Africa" aims to examine the use of low-cost PM sensors as ground data sources for converting satellite AOD retrievals to surface PM2.5. Linear conversion factors relating satellite AOD to surface PM2.5 are calculated. In Pittsburgh, PA, the performance of the low-cost sensors is evaluated compared to traditional regulatory grade monitors, while in Africa, where traditional monitors are lacking, the ability of low-cost sensors to provide satellite AOD conversion factors is examined.

[Figure]

I am recommending the paper undergo major revisions.

General Comments

The majority of the results section focuses on the analysis for the Pittsburgh region. The goal of the paper is to assess the utility of low-cost sensors in deriving satellite AOD conversion factors, however, the results for Pittsburgh seem to suggest that ground monitor data overall performs poorly as a data source for the conversions over the region, at least in terms of correlations. As the authors note, this is likely due to the low concentrations being within the range of signal-to-noise in the sensors. This makes the results less meaningful, because it is difficult to determine whether the results are reflecting the ability of the low-cost sensors to be data sources for the satellite AOD conversion, or whether the results are just overwhelmed by the uncertainties in the measurements, and undermines the authors' conclusions that low-cost sensors perform just as well if not slightly better than the regulatory grade monitors in this region.

The analysis over Africa appears to be more promising, but much less time is spent discussing those results. The authors may be better suited by more prominently pre-senting the analysis over Africa. Low-cost sensor data would provide more benefit over regions such as Africa where the regulatory grade monitors are sparse; there already exist dense regulatory grade monitors over North America, so focusing more on the analysis over Africa would be of greater interest. Describing in detail the comparison of low-cost sensors and regulatory grade monitors in Pittsburgh would make sense if the results were meaningful, as they would provide a meaningful evaluation of the ability of the low-cost sensors to be used to convert satellite AOD in general, but in this case the results seem to suggest the method just doesn't work over Pittsburgh, and does little to provide confidence in the low-cost sensor only analysis over Africa.

Specific Comments

- Several of the figures are difficult to decipher. Figure 2 is difficult to read because the

labels on the y-axis are clustered so close together. Figure 7 is extremely difficult to interpret, because it is hard to see the shades of red. Supplemental figures S6-S9 are very hard to follow and do not help to clarify the methods.

- In addition to Figure S5, the authors should have map plots for each region with the monitor locations over-laid, with a better indicator for the distance between monitors than just latitude and longitude. It is very difficult from Fig S5 to discern where the monitors are positioned throughout the cities, which would provide insight into the results. It is very difficult to tell which monitors are low-cost and which are regulatory without looking extremely closely.

- It is unclear how the satellite AOD and ground monitor data are being sampled; are the authors using pixels co-located to the ground monitor sites, or are they comparing a broader area of AOD to the ground monitor points? Also at which time-scales are the data points being sampled? At satellite-overpass time? This information would have important implications for the results.

- In several instances more "methods" type descriptions are mixed in with the results. Having all methods descriptions in the methods section would make the presentation of the results clearer.

Minor comments:

- Line 70: what is a "good" correlation? No range of values from the studies is given.

- Throughout the manuscript the authors refer to "satellite AOD measurements", when technically they are retrievals and not direct measurements.

- In the introduction the second paragraph on page 3 is confusing. It is structured as though they are discussing studies that use models combing satellite AOD with CTMs to estimate PM2.5, but then all of a sudden they are discussing satellite AOD and ground monitor PM2.5 agreement over Africa.

- When discussing the yearly/monthly conversion factors on page 11, it is unclear

whether the monthly conversion factors are applied on a monthly basis, or if they are calculated on a monthly basis then applied on an annual basis: "the 'monthly' case, data from the previous month are used to develop conversion factors used in the current month; the median performance across months is presented".

———————————————————

---

## Author Response (AR1)

The authors thank the reviewers for their helpful comments. Reviewer comments are reproduced here in red, while our responses are indicated in blue. Where applicable, passages from the manuscript have been reproduced.

We would also like to note a slight change in methodology compared with the initial version of this paper. In the revision, we use AOD at either 470nm or 550nm as independent variables in the regression, whereas before both were used. Because of the high correlation of these two variables, there was little added value to including both, and in fact this led to worse performance for certain datasets with minimal initialization data. By instead using only one of the two variables, the results are generally more robust.

**Reviewer 1**

The goal of this paper is to assess the conversion of satellite AOD values (not measurements)...

References to satellite AOD "measurements" have been modified throughout the paper; we refer to these instead as AOD data or AOD retrievals.

Most of my comments are for PA since the paper focuses on this region.

The study region is very small 0.7 degrees by 0.7 degrees. The paper definitely needs a map of some sort showing the location of the regulation grade monitors and the location of the low cost sensors since I have no idea how close are far away these sensors are!

Additional maps for the ground calibration sites have been included in the supplemental information as Figures S4 through S9, with background maps including local landmark and scale information.

The paper never discusses as to how space-time collocation was done for the ground versus satellite data. The results vary depending upon the width of the time and space windows. The paper also does not provide the slope/intercept values for these linear correlations.

This discussion has been included in Section 2.4 (lines 216-218):

"Satellite AOD data are considered to be collocated in space with data from a ground site when the center of the AOD pixel is within 1 km of the ground site. Data are considered concurrent if the satellite overpass occurs within the hour interval over which ground site data have been averaged to arrive at the hourly-average PM2.5 concentration value used."

Slope and intercept values have been included in the supplemental information, Section S3.1.

The range of annual values in PA was low and the satellite data and the low cost sensors have larger uncertainties in this range and therefore the results may not be robust. Given this backdrop I am not sure how meaningful the PA results are. This is probably the main reason that the correlations are low – Page 11 (Line 325+). Not sure about the usefulness of an offline approach where only a single conversion factor is used. Why report these values when we know that this is not relevant?

In the Pittsburgh area, we are able to analyze the effect of ground monitoring network density, which is not possible with the currently sparsely-monitored African locations. Section 3.2 shows that the satellite AOD to surface PM conversion uncertainty reduces meaningfully for up to about ten low-cost sensors over the 600 square kilometer area, which is useful guidance for future low-cost sensor deployments, including those planned for African cities. Further, the results for Pittsburgh are presented to provide a baseline

and contrast for the results obtained for Sub-Saharan Africa, where the low-cost sensor and satellite data combination is thus seen to be quite valuable. A single conversion factor is used as it represents the simplest and most robust calibration method, while more sophisticated calibrations might be subject to over-fitting to the calibration data sets. Results are presented for the offline approach as a baseline to compare with an online approach, to assess what benefit if any the online approach provides.

Page 11, Line 319. What is the cloud cover for each site and how does it affect annual average AOD? Given some of the issues mentioned above I am not sure that page 12 (line 24-244) conclusion is acceptable. Also given that the linear correlation has so many problems, using satellite data and ground monitors to assess the linear relationship is fraught with uncertainties.

Information on satellite data coverage is included in the supplemental information, Tables S2 and S3. However, since we do not consider long-term average values in this work, we only compare cloud-free AOD to surface PM measurements taken at approximately the same time (i.e. as the hourly average value for the period in which the satellite overpass occurs). Therefore, there will be no issues related to sampling bias for only using data from cloud-free days in these comparisons, as would be the case if we were looking at longer averaging periods.

Although we agree that the methods presented have numerous inherent uncertainties, a major goal of this paper is to assess whether, even with such uncertainties, useful results can be obtained by combining low-cost sensor and satellite data. We find that this is the case, at least in the context of Sub-Saharan Africa where signal-to-noise ratios can be higher and there is very little ground-based monitoring.

In summary, I believe that low cost sensors play an important role for PM2.5 research but unless calibration issues and comparisons with ground monitors of regulation grade are made carefully as a function of space, time, meteorology we cannot be sure how useful the data can be for quantitative monitoring, assessment, and research (e.g. epidemiology). It is also not fair to state that (Page 16, line 482) that using the nearest monitor is better than using satellite data because none of the meteorological factors have been taken into account for estimating PM2.5 from satellite data.

Careful corrections by collocation-based comparison of low-cost PM sensors with regulatory-grade monitors and different methodological approaches in the Pittsburgh context have been the subject of a previous paper (Malings et al. 2019b as cited in the paper, DOI 10.1080/02786826.2019.1623863). We have included, where possible, performance assessments for the low-cost monitors in other contexts, but this is a subject of ongoing work and beyond the scope of this paper. Rather, this paper represents a preliminary attempt to quantify the usefulness of simple linear relationships between AOD and ground PM from low-cost sensors, even taking into account any inherent uncertainties these instruments may have.

We did not mean to assert that the use of nearby sensors was always better than using satellite data in all contexts, but merely within the current high-spatial-density monitoring network in Pittsburgh and the confines of the linear conversion method applied. This statement has been clarified in the text (lines 501-504):

"However, it was found that for Pittsburgh, with a relatively dense low-cost sensor network (median inter-site distance of about 1 km) and low  $PM_{2.5}$  concentrations, use of the nearest

ground measurement sites outperformed the use of satellite AOD data to estimate surface  $PM_{2.5}$  using linear conversions."

**Minor comments**

Wang and Christopher, 2003 – Not Wang, 2003

We apologize for the oversight. This has been corrected.

Some of the references are outdated. E.g. Zhang et al 2009 for correlation coefficients.

This particular reference has been removed in Section 3.1, but has been retained in the Introduction for its value in providing general background information on AOD to surface PM correlations.

**Page 3 : What spatial/temporal scales did Murray et al used**

This paper made use of 12-km spatial scale data at daily temporal resolution. This has been noted in the text (lines 86-88):

"Methods incorporating the outputs of chemical transport models (in this case at lower spatial resolutions of 12 km compared to the 1 km AOD resolution, and at daily temporal resolution) can further improve these results (e.g. Murray et al., 2019)."

**Page 3 : Not all studies find 'anti-correlation' in India.**

Thank you for pointing this out. Since our present work does not cover India, this information is not strictly relevant, and so we no longer reference it in the paper.

**Page 3: Last sentence needs a reference**

Low-cost sensor and reference monitor typical prices are based on manufacturer prices in our experience from the past several years. This has been stated (lines 103-107):

"Low-cost air quality monitors have much lower purchase and operational costs in contrast to traditional or regulatory-grade monitors (Snyder et al., 2013; Mead et al., 2013). For example, a lower-cost multi-pollutant monitor (measuring gases and PM) costs a few thousand US dollars; single-pollutant PM sensors can cost just a few hundred US dollars. A comparable multi-pollutant suite of traditional air quality monitoring instruments would cost a hundred thousand US dollars or more; a regulatory-grade PM monitor can cost tens of thousands of US dollar (based on recent manufacturer quotations)."

Page 3: The cloud cover problems needs to be addressed and referenced. Christopher & Gupta (2010) Satellite Remote Sensing of Particulate Matter Air Quality: The Cloud-Cover Problem, Journal of the Air & Waste Management Association, 60:5, 596-602, DOI: 10.3155/1047-3289.60.5.596

This reference has been added to the introduction section (lines 67-70):

"Cloud cover also makes AOD retrievals impossible; the frequency of cloudy days in an area can therefore make it difficult to establish reliable relationships between AOD and surface PM, although this is not likely to be a concern for the continental US (Christopher and Gupta, 2010; Belle et al., 2017)."

The cloud cover problem can be important for long-term averages. As noted previously, cloud cover is not an issue in our comparisons because we focus on hourly data during cloud-free periods (lines 218-223):

"As we compare data from individual satellite passes directly to temporally collocated ground site data, we do not need to consider (as would be essential for long-term averages) the potential impact of the fraction of time where satellite measures are missing (due to cloud cover or other factors). Likewise, we do not consider the biases associated with the fact that satellite passes occur at certain times of day (required when comparing with daily-averaged ground monitoring data) since here we only compare AOD to surface PM2.5 during the same hour when the satellite pass occurs."

Page 4: Errors cannot average out and it depends on the range of PM2.5 values and a host of other factors.

This was a conjecture as to possible future applications of satellite and low-cost sensor data. The sentence has been removed.

Section 2.1.1 to 2.1.3 belongs in a Table rather than a few sentences of text

We thank the reviewer for this suggestion. The information presented in these sections, as well as basic details of the study areas, have been presented in Tables 1 and 2.

Page 5: Line 1 : Here not hare

This has been corrected.

Page 7 says 'as summarized in 2.1.4' but 2.1.4 does not describe calibration in any detail. Erroneous data screening for negative values is easy but doing this manually for the entire low cost network is not possible.

A full presentation of the calibration methods is beyond the scope of this work, and is more fully covered in the cited publication (Malings et al., 2019b). While it is true that manual error detection and elimination for a large network of sensors is difficult, it can be aided through the use of certain automatic processes. While we seek to present data that has been calibrated and validated to the best of our abilities, we acknowledge that fool-proof error detection and correction is not possible. Such errors are a source of uncertainty in the present work, and one of our major goals with this paper is to demonstrate and quantify the extent to which low-cost sensor data, even with these uncertainties, can provide additional information to support the conversion of AOD to surface PM2.5.

Additional details have been provided in the text (Section 2.1, lines 160-170):

"Collected data are down-averaged from their device-specific collection frequencies to a common hourly timescale. Erroneous data identified either automatically (e.g. negative concentration values or unrealistically high or low values) or manually (e.g. devices exhibiting abnormal performance characteristics identified during periodic inspections) are removed. To correct for particle hygroscopic growth effects (i.e. the impact of ambient humidity on the PM mass as measured by the low-cost sensors), previously developed calibration methods (Malings et al., 2019b) were implemented for the NPM and PA-II sensors. Briefly, first, a hygroscopic growth factor is computed using the local humidity and temperature as measured by the low-cost monitor itself, along with an average or typical particle composition. Then, a linear correction is applied to the data based on past collocations with regulatory-grade monitoring instruments. Utilizing these methods, the uncertainties on hourly average  $PM_{2.5}$  concentration are about 4  $\mu$ g/m3 (Malings et al., 2019b). For the Alphasense OPC sensors, raw bin count numbers were integrated to produce a new concentration estimate for  $PM_{2.5}$ , and a similar relative humidity correction was applied (Di Antonio et al., 2018)."

Page 6: Line 180-183 says the data are scaled for workdays and non work days. This type of scaling may work for this study but how about other regions?

Indeed, different scaling factors may be necessary in other regions, and this is the subject of ongoing research on the generalizability of low-cost sensor calibration approaches across the vast continent of Africa. For the purposes of this paper, we seek to use data from low-cost sensors which represent the best available practices in each instance. Therefore, we have included scaling factors in Rwanda based on applicable local comparisons and calibration. Since we are using linear methods, the presence or absence of linear scaling factors that are equally applied to both training and testing sets of low-cost sensor data should not influence the assessment of the methodology.

Page 8: The satellite data needs some description with a proper journal reference. Briefly, how was AOD retrieved, what are the uncertainties, how much cloud cover for the analysis, what quality flags were used, etc.

A more complete description of the satellite data has been provided in Section 2.4 (lines 205-223):

"The satellite data product used in this paper is the MODIS MCD19A2v006 dataset (Lyapustin and Wang, 2018) available through NASA's Earth Data Portal (earthdata.nasa.gov). This dataset consists of AOD information for the 470nm and 550nm wavelengths from the MODIS system, processed using the Multi-angle Implementation of Atmospheric Correction (MAIAC) algorithm, and presented at 1 km pixel resolution for every overpass of either the Aqua or Terra satellites (Lyapustin et al., 2011a, 2011b, 2012, 2018). This represents a Level 2 data product, meaning that it includes geophysical variables derived from raw satellite data, but has not yet been transformed to a new temporal or spatial resolution, as is the case for data derived from multiple satellite passes, e.g. monthly average AOD data. Data from identified cloudy pixels are masked as part of the data product; possible misidentification of cloudy pixels is one source of error in relating surface PM2.5 and AOD. As per recommendations in the User Guide for this dataset, only data matching "best quality" quality assurance criteria are used. This dataset was chosen as it represents the highest possible spatial and temporal resolution for AOD, thus providing the most points for comparison with the high spatio-temporal resolution low-cost monitor data.

Satellite AOD data are considered to be collocated in space with data from a ground site when the center of the AOD pixel is within 1 km of the ground site. Data are considered concurrent if the satellite overpass occurs within the hour interval over which ground site data have been averaged to arrive at the hourly-average PM2.5 concentration value used. As we compare data from individual satellite passes directly to temporally collocated ground site data, we do not need to consider (as would be essential for long-term averages) the potential impact of the fraction of time where satellite measures are missing (due to cloud cover or other factors). Likewise, we do not consider the biases associated with the fact that satellite passes occur at certain times of day

(required when comparing with daily-averaged ground monitoring data) since here we only compare AOD to surface PM2.5 during the same hour when the satellite pass occurs."

Additional details on the cloud cover and uncertainty analysis are included in the supplemental information, Tables S2 and S3.

**Reviewer 2**

**General Comments**

The majority of the results section focuses on the analysis for the Pittsburgh region. The goal of the paper is to assess the utility of low-cost sensors in deriving satellite AOD conversion factors, however, the results for Pittsburgh seem to suggest that ground monitor data overall performs poorly as a data source for the conversions over the region, at least in terms of correlations. As the authors note, this is likely due to the low concentrations being within the range of signal-to-noise in the sensors. This makes the results less meaningful, because it is difficult to determine whether the results are reflecting the ability of the lowcost sensors to be data sources for the satellite AOD conversion, or whether the results are just overwhelmed by the uncertainties in the measurements, and undermines the authors' conclusions that low-cost sensors perform just as well if not slightly better than the regulatory grade monitors in this region.

One of the major motivations for including the results from Pittsburgh is to present a baseline case for a densely monitored (with both regulatory and low-cost monitors) region in order to contrast with results from more sparsely monitored locations in Rwanda and elsewhere. In particular, although we agree that overall performance of the satellite AOD to ground  $PM_{2.5}$  conversion is rather poor in the conditions of Pittsburgh, it is at least consistent for both ground data sources (regulatory reference instruments and low-cost monitors). Note that the typical  $PM_{2.5}$  concentrations in Pittsburgh (an inter-quartile range of 6 to 12 µg/m3) are still above the hourly-average measurement uncertainty (3 to 4 µg/m3) of the low-cost sensors. Considering the reasonable agreement between low-cost and regulatory-grade monitors identified in previous work, together with the observation from this work that performance is not noticeably disadvantaged by the substitution of regulatory-grade for low-cost monitors, we believe it is reasonable to assume that most of the poor performance of the satellite AOD to ground  $PM_{2.5}$  conversion is due to the inherent difficulties of this problem and the low-concentration regime of Pittsburgh, rather than the data quality of the ground source. We have restated the conclusion based on our comparative analysis of low-cost and regulatory-grade instruments in Pittsburgh to better emphasize this (lines 389-396):

"In all cases, performances using low-cost sensor data are comparable to that of the same conversion approaches utilizing the regulatory-grade instruments. Note that the low-cost monitors used here have been carefully corrected by collocation with regulatory-grade monitors (Malings et al., 2019b) which accounts for known artefacts with low-cost sensors. Thus, there is no evidence from this analysis of any inherent disadvantage to the use of carefully corrected low-cost sensors to provide ground data as compared to more traditional instruments. Rather, based on these results, any additional uncertainty due to data quality differences between low-cost sensors and regulatory-grade instruments are seen to be negligible compared to the difficulties associated with relating satellite AOD to surface-level PM2.5, and therefore have had no systematic impact on the performance of the assessed linear conversion method, at least for this study area."

The analysis over Africa appears to be more promising, but much less time is spent discussing those results. The authors may be better suited by more prominently presenting the analysis over Africa. Low-cost sensor data would provide more benefit over regions such as Africa where the regulatory grade monitors are sparse; there already exist dense regulatory grade monitors over North America, so focusing more on the analysis over Africa would be of greater interest. Describing in detail the comparison of low-cost sensors and regulatory grade monitors in Pittsburgh would make sense if the results were meaningful, as they would provide a meaningful evaluation of the ability of the low-cost sensors to be used to convert satellite AOD in general, but in this case the results seem to suggest the method just doesn't work over Pittsburgh, and does little to provide confidence in the low-cost sensor only analysis over Africa.

We thank the reviewer for recognizing the potential benefit of low-cost sensors for Africa. This is a point we seek to make and support quantitatively through the results presented in this paper. We have expanded our discussion of results in Africa to increase the relative emphasis placed on these results. We have also reorganized the paper somewhat and restructured the discussion of the results, including a new figure related to this discussion (Figure 6) to better emphasize the relative significance and importance of the results for Africa. However, we feel it is also important to present the "weaker" results for Pittsburgh as a basis of comparison for the more promising results for sub-Saharan Africa. Furthermore, the analysis of the potential benefits of high spatial density low-cost sensor networks (the "how many sensors are needed" question) can only be performed using the Pittsburgh data, where such a network has been operational since 2016.

**Specific Comments**

- Several of the figures are difficult to decipher. Figure 2 is difficult to read because the labels on the yaxis are clustered so close together. Figure 7 is extremely difficult to interpret, because it is hard to see the shades of red. Supplemental figures S6-S9 are very hard to follow and do not help to clarify the methods.

The vertical spacing of Figure 2 has been increased. The color scale of Figure 7 has been changed to improve interpretability. Numerical values corresponding to these colors have also been provided in the supplemental information (Table S8). Supplemental Figures S6 to S9 were augmented with a more detailed narrative description of the methods (including new figures, with all figures now numbered S11 to S18), which we believe makes these points more clearly.

- In addition to Figure S5, the authors should have map plots for each region with the monitor locations over-laid, with a better indicator for the distance between monitors than just latitude and longitude. It is very difficult from Fig S5 to discern where the monitors are positioned throughout the cities, which would provide insight into the results. It is very difficult to tell which monitors are low-cost and which are regulatory without looking extremely closely.

Map plots depicting the locations of the monitors have been included in the supplemental information as Figures S4 through S9. The markers are much larger and are overlaid on geographical maps which should help better illustrate the monitor locations.

- It is unclear how the satellite AOD and ground monitor data are being sampled; are the authors using pixels co-located to the ground monitor sites, or are they comparing a broader area of AOD to the ground

monitor points? Also at which time-scales are the data points being sampled? At satellite-overpass time? This information would have important implications for the results.

A more complete description of the sampling method has been provided in Section 2.4 (lines 205-223):

"The satellite data product used in this paper is the MODIS MCD19A2v006 dataset (Lyapustin and Wang, 2018) available through NASA's Earth Data Portal (earthdata.nasa.gov). This dataset consists of AOD information for the 470nm and 550nm wavelengths from the MODIS system, processed using the Multi-angle Implementation of Atmospheric Correction (MAIAC) algorithm, and presented at 1 km pixel resolution for every overpass of either the Aqua or Terra satellites (Lyapustin et al., 2011a, 2011b, 2012, 2018). This represents a Level 2 data product, meaning that it includes geophysical variables derived from raw satellite data, but has not yet been transformed to a new temporal or spatial resolution, as is the case for data derived from multiple satellite passes, e.g. monthly average AOD data. Data from identified cloudy pixels are masked as part of the data product; possible misidentification of cloudy pixels is one source of error in relating surface PM2.5 and AOD. As per recommendations in the User Guide for this dataset, only data matching "best quality" quality assurance criteria are used. This dataset was chosen as it represents the highest possible spatial and temporal resolution for AOD, thus providing the most points for comparison with the high spatio-temporal resolution low-cost monitor data.

Satellite AOD data are considered to be collocated in space with data from a ground site when the center of the AOD pixel is within 1 km of the ground site. Data are considered concurrent if the satellite overpass occurs within the hour interval over which ground site data have been averaged to arrive at the hourly-average PM2.5 concentration value used. As we compare data from individual satellite passes directly to temporally collocated ground site data, we do not need to consider (as would be essential for long-term averages) the potential impact of the fraction of time where satellite measures are missing (due to cloud cover or other factors). Likewise, we do not consider the biases associated with the fact that satellite passes occur at certain times of day (required when comparing with daily-averaged ground monitoring data) since here we only compare AOD to surface PM2.5 during the same hour when the satellite pass occurs."

- In several instances more "methods" type descriptions are mixed in with the results. Having all methods descriptions in the methods section would make the presentation of the results clearer.

We thank the reviewer for this suggestion. These descriptions have been moved into their own subsection (2.6) within the "Methods" section.

Minor comments:

- Line 70: what is a "good" correlation? No range of values from the studies is given.

A representative value from the reference has been provided (lines 72-75):

"Nevertheless, early examinations of AOD data from the Moderate Resolution Imaging Spectroradiometer (MODIS) instrument, launched aboard the Terra and Aqua satellites in 1999 and 2002, showed good correlation (e.g. correlation coefficient r about 0.7 for Jefferson County, Alabama in 2002) with surface PM2.5 concentrations in the United States, although these relationships varied from region to region (Wang and Christopher, 2003; Engel-Cox et al., 2004)."

- Throughout the manuscript the authors refer to "satellite AOD measurements", when technically they are retrievals and not direct measurements.

References to satellite AOD "measurements" have been modified throughout the paper. We now refer to these as AOD data or AOD retrievals.

- In the introduction the second paragraph on page 3 is confusing. It is structured as though they are discussing studies that use models combing satellite AOD with CTMs to estimate PM2.5, but then all of a sudden they are discussing satellite AOD and ground monitor PM2.5 agreement over Africa.

This paragraph has been split into two to better present these different topics.

- When discussing the yearly/monthly conversion factors on page 11, it is unclear whether the monthly conversion factors are applied on a monthly basis, or if they are calculated on a monthly basis then applied on an annual basis: "the 'monthly' case, data from the previous month are used to develop conversion factors used in the current month; the median performance across months is presented".

These factors are applied on a monthly basis. This has been clarified in the text (lines 295-299):

[revised manuscript text omitted]

**2.172.4 Satellite data**

295

The satellite data product used in this paper is the MODIS MCD19A2v006 dataset (Lyapustin and Wang, 2018) available through NASA's Earth Data Portal (earthdata.nasa.gov). This dataset consists of AOD information for the 470nm and 550nm wavelengths from the MODIS system, processed using the Multi-angle Implementation of Atmospheric Correction (MAIAC) algorithm, and presented at 1 km-kilometer pixel resolution for every overpass of either the Aqua or Terra satellites (Lyapustin et al., 2011a, 2011b, 2012, 2018). This represents a Level 2 data product, meaning that it includes geophysical variables derived from raw satellite data, but has not vet been transformed to a new temporal or spatial

resolution, as is the case for data derived from multiple satellite passes, e.g. monthly average AOD data. Data from identified 305 cloudy pixels is are masked as part of the data product; possible misidentification of cloudy pixels is one source of error in relating surface PM2.5 and AOD. As per recommendations in the User Guide for this dataset, only data matching "best

- quality" quality assurance criteria are used. This dataset was chosen as it represents the highest possible spatial and temporal resolution for AOD, thus providing the most points for comparison with the high spatio-temporal resolution low-cost monitor data.
- 310 Satellite AOD data are considered to be collocated in space with data from a ground site when the center of the AOD pixel is within 1 km of the ground site. Data are considered concurrent if the satellite overpass occurs within the hour interval over which ground site data have been averaged to arrive at the hourly-average PM2.5 concentration value used. As we compare data from individual satellite passes directly to temporally collocated ground site data, we do not need to consider (as would be essential for long-term averages) the potential impact of the fraction of time where satellite measures are missing (due to
- 315 cloud cover or other factors). Likewise, we do not consider the biases associated with the fact that satellite passes occur at certain times of day (required when comparing with daily-averaged ground monitoring data) since here we only compare AOD to surface PM2.5 during the same hour when the satellite pass occurs.

**2.182.5 Conversion Methods for satellite AOD-data**

A linear regression approach is used to establish relationships between satellite AOD and surface-level PM2.5. Let  $y_{i,t}$  denote the ground-level PM2.5 measurement at location *i* and time *t*, and let  $x_{i,t}$  represent the vector of satellite AOD measurements (i.e., a vector combining the AOD measurements at 470nm and or 550nm wavelengths, together with a "placeholder" constant of one to allow fitting of affine functions) corresponding to location *i* and time *t*. For this paper we present results using AOD at 550nm; results for AOD at 470nm are similar and are included in the supplemental information (Sect. S3.2). The total set of ground measurement sites in an area, *S*, is partitioned into two disjoint sub-sets. Subset  $S_{in}$  represents the Formatted: Subscript

**Formatted: Subscript**

- 325 sites used to establish the linear relationship between AOD and surface PM2.5 concentrations. The remainder of sites, in the subset  $S_{ap}$ , are used for the application, i.e., to serve as an independent set to evaluate the performance of the linear relationship established from the  $S_{in}$  sites. Likewise, the time domain *T* is partitioned into initialization phase  $T_{in}$ , during which linear relationships are established, and application phase  $T_{ap}$ , during which these relationships are applied and evaluated.
- 330 Linear relationships are determined as follows. First, satellite AOD data and surface PM2.5 monitor data from the  $S_{in}$  sites during the  $T_{in}$  phase were are collected together:

$$X_{\text{in}} = \{x_{i,t}\} \quad \forall i \in S_{\text{in}}, t \in T_{\text{in}}, \tag{1}$$

A linear relationship is established between these, defined by parameters  $\beta_{in}$ , using classical least-squares linear regression (e.g., Goldberger, 1980):

335
$$\beta_{\rm in} = (X_{\rm in}^{\rm T} X_{\rm in})^{-1} X_{\rm in}^{\rm T} Y_{\rm in},$$
 (2)

The covariance matrix of the parameters,  $\Sigma_{\beta_{in}}$ , is also obtained:

340

$$\Sigma_{\beta_{in}} = \frac{(Y_{in} - X_{in}\beta_{in})^{\mathrm{T}}(Y_{in} - X_{in}\beta_{in})}{\operatorname{length}(Y_{in}) - \operatorname{length}(\beta_{in})} \left( X_{in}^{\mathrm{T}} X_{in} \right)^{-1},$$
(3)

where length(·) is a function returning the number of elements in the input. During the application phase, the linear relationship can be used to estimate the surface PM2.5 concentration at location *i* and time *t*,  $\hat{y}_{i,t,prior}$ , from the satellite AOD data corresponding to that location and time:

$$\hat{y}_{i,t,\text{prior}} = x_{i,t} \beta_{\text{in}},\tag{4}$$

The above procedure constitutes an offline or (in Bayesian terminology) prior conversion, i.e., it uses data collected during the initialization phase to define a single conversion factor which that is applied throughout the application phase. An online, dynamic, or (in Bayesian terminology) posterior approach can also be adopted, in which this relationship is modified as additional data are available. This approach has been proposed by Lee et al. (2011) and evaluated by Han et al. (2018), and allows for the potentially time-varying relationship between satellite AOD and surface PM2.5 concentration to be accounted for. In the online approach, for a time *t* during the application phase, a new data set consisting of *Y*in,t and *X*in,t is created by combining all data available from the *S*in ground sites together with satellite AOD data for that time:

$$X_{\mathrm{in},t} = \left\{ x_{i,t} \right\} \quad Y_{\mathrm{in},t} = \left\{ y_{i,t} \right\} \quad \forall \ i \in S_{\mathrm{
[revised manuscript text omitted]

  - case of Pittsburgh) about 15-1 monitors across per 600 square kilometerskm2

**3.3 Comparison of AOD-based surface PM2.5 to measurements from a dense ground network**

Performance of both the nearest monitor method and the satellite AOD conversion method are assessed for Pittsburgh in Fig. *A*. It should be noted that all available ground sites have been used for conversion factor initialization in this section, versus a

| Field | Code Changed            |
|-------|-------------------------|
| Field | Code Changed            |
|       |                         |
| Form  | atted: Font color: Auto |
| Form  | atted: Font color: Auto |
| Form  | atted: Font color: Auto |

| -{ | Formatted: Font color: Auto |
|----|-----------------------------|
| -{ | Formatted: Font color: Auto |
| ſ  | Formatted: Font color: Auto |

| Formatted: | Superscript     |
|------------|-----------------|
| Formatted: | Font color: Red |

**Field Code Changed**

limited subset of these in Sect. 3.1, leading to improved performance of this method following the trend noted in Sect. 3.2. Im 575 this section, we assess the benefits of combining satellite AOD and ground-based sensor data, as compared to using groundbased sensor data alone. For this assessment, we compare estimates of surface PM2.5-derived from satellite AOD data, using the methods presented previously in this paper, with estimates based on the surface PM22-measurements alone, which we denote as "nearest monitor" estimates. For this estimation, we make use of a locally constant or naïve interpolation, in which the surface PM22-estimate for a given time and location is the same as the measurement of the nearest available ground 580 monitor (i.e., one of the ground monitors used for establishing conversion factors for the satellite AOD data) at that time:

 $\hat{y}_{i.t.nearest} = y_{i.t.}$  s.t.  $j = \operatorname{argmin}_{k \in S_{i-1}} \operatorname{dist}(i, k)_{\overline{i}}$

[revised manuscript text omitted]

**Field Code Changed**

**765 Competing Interests**

The authors declare that they have no conflict of interest.

**Acknowledgements**

This work benefited from State assistance managed by the National Research Agency under the "Programme d'Investissements d'Avenir" under the reference "ANR-18-MPGA-0011" ("Make our planet great again" initiative).
Additional funding for this study was providedMeasurements in Pittsburgh were funded by the Environmental Protection Agency (Assistance Agreement Nos. RD83587301 and 83628601); and the Heinz Endowment Fund (Grants E2375 and E3145). Measurements in Rwanda were supported by the College of Engineering, the Department of Engineering and Public Policy, and the Department of Mechanical Engineering at Carnegie Mellon University via discretionary funding support for Paulina Jaramillo and Allen Robinson. AG and AB acknowledge support from NSF Award CNH 1923568 to establish measurement sites in Malawi. The authors would like to thank Jimmy Gasore, Valérien Baharane, Abdou Safari Kagabo, Eric Lipsky, Sriniwasa P.N. Kumar, Provat Saha, Yuge Shi, Naomi Zimmerman, Rebecca Tanzer, and S. Rose Eilenberg for assistance with instrument setup and operation. We also acknowledge the contributions of former Columbia University undergraduate student Karen Xia, supported by a Columbia Earth Institute Research Internship, for data analysis which is

presented in the supplemental information of this document, Fig. S1. Finally, the authors would like to thank Juan Cuesta 780 and Jiayu Li for discussions and advice related to satellite data usage.

Field Code Changed

[revised manuscript text omitted]

---

## Author Response (AR3)

We thank the editor for his comments and suggestions, which are reproduced below in red. Our responses are presented in blue.

Section 2.2: Your Supplement contains maps showing the locations of the various monitors in each region. I think that it would be beneficial to move Figure S4 (the map for Pittsburgh, PA) to the main paper file. This is the example "dense" network and it will be useful for the reader to see the distribution of sites (regulatory and low-cost) more clearly. Unfortunately, I know a lot of people won't look at the Supplement. The other maps of this type are probably fine to keep in the Supplement because there are far fewer monitors in them.

This figure has been moved to the main paper as Fig. 2.

Line 157: There is a missing section reference here ("(see Sect. Error! Reference source not found.)." appears in the manuscript). Please check and correct.

The section in question was removed during the previous revisions. This reference now indicates Table 2, where the relevant information is now presented.

Lines 206-208: "This represents a Level 2 data product, meaning that it includes geophysical variables derived from raw satellite data, but has not yet been transformed to a new temporal or spatial resolution, as is the case for data derived from multiple satellite passes, e.g. monthly average AOD data." This is not quite true, because MAIAC is in fact transformed from the (irregular) satellite swath grid to an Earth-referenced grid. Also, many level 2 products also coarsen their data compared to level 1 resolution. So I'd streamline this sentence to say something like "This represents a Level 2 data product, meaning that it is provided as a snapshot at the satellite overpass times and has not been aggregated to coarser (e.g. monthly) temporal resolution." That seems to be the main aspect you were trying to capture with this text.

This text has been adjusted as follows (lines 206-207):

> "This represents a Level 2 data product, meaning that it includes geophysical variables derived from raw satellite data at each overpass time and has not been aggregated to a coarser (e.g. monthly) temporal resolution."

Section 3.5: I did not find an explicit definition of the seasons used in the paper (here, earlier, or the Figure 7 caption). This should be added somewhere.

The seasons are defined in the Supplemental Information, Table S1. This has been indicated in the text in this section (lines 450-451):

> "Fig. 8 presents the median performance metrics across all sites in either Pittsburgh or Rwanda for each combination of initialization and application season. Seasonal definitions are provided in the supplemental information, Table S1."

Figure 8, and associated discussion: it would be useful to provide uncertainty estimates for these metrics like correlation coefficient. This could be done by e.g. the bootstrap method (sampling the data with replacement many times and showing the resulting standard deviation of the estimates). It is not clear to me how distinct some of these results are from one another given the different data volumes at different locations. This should then be borne in mind when making conclusions.

Uncertainty estimates have been provided via bootstrapping as you suggested. This has been indicated in the text (lines 357-359):

> "Uncertainty estimates for the performance of this approach at each site are obtained via bootstrap resampling of the times with valid coincident satellite and ground data, with 100 random bootstrap samples being used to obtain the uncertainty estimates."

The resulting figure has also been updated (this is now Figure 9):

[Figure]

Figure 9: Comparison of performance metrics (a: correlation, b: CvMAE, c: MNB) for surface PM2.5 estimated from satellite AOD data across multiple sites in SSA. The conversion factor is developed at a central site in Kigali, Rwanda; the distances of each testing site to this central site are given. Performances are assessed for all data collected at the given sites, using the prior conversion factor only. Note that performance in Kampala and Addis Ababa is assessed using traditional reference monitors (indicated by ∗), while performance at the other sites reflects low-cost sensor data (indicated by ●). Error bars denote the interquartile range of metric estimates obtained via bootstrap resampling (for most cases of the mean-normalized bias, this range is smaller than the marker size).

Overall, inter-quartile ranges of the estimates tend to be smaller than the differences between estimates for different sites, so we believe that the results as presented and discussed in the paper are robust to these uncertainties.

Lines 516-518: this is true for polar-orbiting sensors, although less so for geostationary platforms. However, until recently the geo sensors haven't been as capable as polar-orbiting ones so these data haven't been so widely-used for AOD retrieval. A MAIAC-based approach has been used before, though, e.g. Zhang et al (2011: https://www.atmos-chem-phys.net/11/11977/2011/ and 2013: https://www.atmos-meas-tech.net/6/471/2013/ )

Thank you for this information. The discussion of geostationary satellites has been expanded (lines 518-525):

> "On a related point, satellite data (at least, for most of the world using current polar-orbiting platforms) cannot provide diurnal concentration profiles, instead presenting a "snapshot" of concentrations for a wide spatial domain but only for a specific time of day. Ground-based continuous monitoring, even with low-cost sensors, will still be essential where there is no coverage with geostationary platforms which provide continuous (for daytime only) retrievals (Judd et al., 2018; She et al., 2020). Past work has made use of AOD retrievals from GOES geostationary satellites for North America (Zhang et al., 2011, 2013). New geostationary satellites are planned for coverage of North America (the TEMPO satellite mission), Europe (Sentinel 4), and East Asia (GEMS); unfortunately, there are no current plans for coverage of Africa by similar satellites."

I'd note also that your Figures using Google Maps for the background require attribution of the data sources more directly – see the Google Maps reuse guidelines at https://www.google.com/permissions/geoguidelines/ and, more specifically, https://www.google.com/permissions/geoguidelines/attr-guide/ .

Appropriate attributions according to these guidelines have been added to the captions of each figure, for example (caption of Figure 2):

> "Background map obtained from maps.google.com, map data ©2020 Google."

[revised manuscript text omitted]